# Base editing-mediated one-step inactivation of the *Dnmt* gene family reveals critical roles of DNA methylation during mouse gastrulation

Qing Li[1,9], Jiansen Lu[2,9], Xidi Yin[1,9], Yunjian Chang[3,9], Chao Wang[3,9], Meng Yan[4,9], Li Feng[5,6,7], Yanbo Cheng[8], Yun Gao[2], Beiying Xu[3], Yao Zhang[3], Yingyi Wang[8], Guizhong Cui[1], Luang Xu[3], Yidi Sun[5], Rong Zeng[6], Yixue Li[7], Naihe Jing[1], Guo-Liang Xu[3] ✉, Ligang Wu[3] ✉, Fuchou Tang[2] ✉ & Jinsong Li[1,4,8] ✉

During embryo development, DNA methylation is established by DNMT3A/3B and subsequently maintained by DNMT1. While much research has been done in this field, the functional significance of DNA methylation in embryogenesis remains unknown. Here, we establish a system of simultaneous inactivation of multiple endogenous genes in zygotes through screening for base editors that can efficiently introduce a stop codon. Embryos with mutations in *Dnmts* and/ or *Tets* can be generated in one step with IMGZ. *Dnmt*-null embryos display gastrulation failure at E7.5. Interestingly, although DNA methylation is absent, gastrulation-related pathways are down-regulated in *Dnmt*-null embryos. Moreover, DNMT1, DNMT3A, and DNMT3B are critical for gastrulation, and their functions are independent of TET proteins. Hypermethylation can be sustained by either DNMT1 or DNMT3A/3B at some promoters, which are related to the suppression of miRNAs. The introduction of a single mutant allele of six miRNAs and paternal *IG*-DMR partially restores primitive streak elongation in *Dnmt*-null embryos. Thus, our results unveil an epigenetic correlation between promoter methylation and suppression of miRNA expression for gastrulation and demonstrate that IMGZ can accelerate deciphering the functions of multiple genes in vivo.

DNA methylation of the fifth carbon of cytosines (5mC) is dynamically erased and re-established during mouse pre- and post-implantation development[1–3]. After fertilization, global 5mC in sperm and oocytes established during gametogenesis is gradually erased in the resultant embryos through TET-mediated active demethylation and DNA replication-mediated passive demethylation, reaching a low point at the blastocyst stage[4,5]. Notably, de novo DNA methylation by DNMT proteins also occurs amid global demethylation and is subjected to TET-mediated DNA demethylation in zygotes[6]. Subsequently, de novo methylation mediated by DNMT3A and DNMT3B occurs, and the developing embryo regains adult levels of methylation during gastrulation, which then preserves through cell divisions by DNMT1 and determines the lineage specification[7,8]. Mouse embryos carrying DNMT1 or DNMT3A/3B mutants have been generated and displayed embryonic lethality at mid-gestation, demonstrating their essential roles in early embryonic development[7,8]. However, although *Dnmt*-null

ESCs and trophoblastic stem cells lacking DNA methylation have been generated and have shown normal self-renewal ability[9,10], *Dnmt*-null embryos have not been achieved, largely owing to the difficulties in establishing the complex breeding system using germline-specific conditional knockout parents. Therefore, the developmental functions of DNA methylation in mouse embryogenesis remain to be determined.

Recently, CRISPR-Cas9-mediated base editing (BE) has been established by combining Cas9 nickase with cytosine deaminase enzyme and uracil glycosylase inhibitor that enables genome rewriting at specific sites through C>T or G>A base replacement[11]. Compared with conventional CRISPR-Cas9-mediated genome editing, the mutations caused by BEs are more precise and controllable[12]. Meanwhile, BE system is an ideal tool to silence genes by introducing premature stop codons (CAA, CAG, CGA, TGG to TAA, TAG, TGA, TAA, respectively) in cultured cells[13,14]. Moreover, BEs can disrupt the targeted genes in the resultant mice through direct injection into zygotes[15,16]. Recently, Liu et al. reported that zygote injection of the BE3 could be employed to produce triple gene knock-out mice[17]. We thus propose that we might use BE3 to investigate the developmental functions of DNA methylation through the simultaneous inactivation of *Dnmt1*, *Dnmt3a*, and *Dnmt3b*. However, two sgRNAs per gene have been used in their system, leading to complex genotypes in the resultant embryos[17]. Meanwhile, since several different BE systems have been developed in past years[18,19], it is necessary to reveal the most efficient one for producing embryos with multiple knockout genes in one step by comparison studies.

In this work, we screen for the most efficient cytosine base editors from six available systems, and identified that hA3A-eBE3-Y130F can efficiently introduce a stop codon at the targeted site through zygote injection. We next establish an IMGZ (inactivation of multiple genes in zygotes) system based on hA3A-eBE3-Y130F to efficiently generate embryos with multiple mutant genes between *Tet* and *Dnmt* family genes. With these mutant embryos, we discover the critical roles of *Dnmt* family genes on embryo gastrulation by regulating the dosage of miRNAs, which are independent of TET proteins.

## Results

### Screening for a BE with the highest mutation rate in zygotes

To efficiently induce stop-codon mutations in mouse embryos, we sought to identify CRISPR-based base editor (BE) with the highest efficiency from six available systems, including BE3, Gam-BE4, hA3A-BE3-Y130F, hA3A-eBE3-Y130F, X-BE3, and X-BE4 (Supplementary Fig. 1a)[20–22]. We first injected mRNA of one of the BE systems and an sgRNA targeting the *Oct4-EGFP* transgene into zygotes to introduce a premature stop codon in *EGFP* transgene, according to an established protocol (Fig. 1a, b)[16]. The results showed that hA3A-eBE3-Y130F[22] led to the most efficient C to T conversion (~90%) in the resultant blastocysts among all tested BE systems, with a higher ratio of homozygous mutation and fewer side-products (Fig. 1c–e; Supplementary Fig. 1b and Supplementary Data 1). We also tested these BEs at an endogenous gene site of crystallin gamma C (*Crygc*) and observed similar results (Supplementary Fig. 1c-g and Supplementary Data 1). We further optimized the injection condition and found that 100 ng/μl of hA3A-eBE3-Y130F and sgRNA respectively resulted in the highest editing efficiency and fewer side effects (Supplementary Fig. 1h-k and Supplementary Data 1). Finally, GOIT (genome-wide off-target analysis by two-cell embryo injection)[23] analysis of three endogenous genes (*Tyr*, *Crygc*, and *Dnmt3a*) indicated that hA3A-eBE3-Y130F induced minor and random off-target mutations caused by deaminase enzyme activity (Fig. 1f–h; Supplementary Fig. 2, 3 and Supplementary Data 2) as other BE systems reported previously[23,24].

### One-step production of embryos with multiple mutated genes

We next attempted to establish a protocol based on hA3A-eBE3-Y130F to induce the inactivation of multiple endogenous genes in zygotes in one step (IMGZ). Previous studies have shown that sgRNA per se determines the editing efficiency[25,26]. To obtain sgRNAs with a high editing efficiency, we first developed a website (named Base-Editor) to design suitable sgRNAs for introducing premature stop codons in a targeted gene (Supplementary Fig. 4a). We next tested the efficiency of IMGZ by simultaneous inactivation of *Tet* family genes in zygotes (Fig. 2a). Three candidate sgRNAs for each gene that may introduce a stop-codon mutation and induce minimal off-target effect predicted by Base-Editor were selected for zygote injection (Supplementary Fig. 4b). The injected zygotes developed to the blastocyst stage, which were then used for mutation efficiency analysis (Supplementary Fig. 4c-e). The most efficient sgRNAs (i.e., Tet1-sg3, Tet2-sg1, and Tet3-sg2, respectively) were then mixed for zygote injection (Fig. 2a, b and Supplementary Data 4).

As expected, injected zygotes developed into blastocysts normally (Supplementary Fig. 4f). Strikingly, Sanger and Blunting sequencing analyses showed that 9 of 16 tested blastocysts (56.3%) carried homozygous premature mutation at the three targeted sites (thus termed *Tet*-TKO embryos) (Fig. 2c and Supplementary Fig. 4g). The Next-Generation Sequencing (NGS) of E8.5 *Tet*-TKO embryos further confirmed the high C-T conversion efficiency in the resultant embryos (Fig. 2d and Supplementary Data 5). Liquid chromatography-mass spectrometry (LC-MS) showed undetectable 5-hydroxymethylcytosine (5hmC) signals in *Tet*-TKO embryos (Fig. 2e), indicating that premature stop mutants of *Tet1/2/3* disrupted the activity of dioxygenases (Fig. 2f and Supplementary Data 3). Interestingly, we found that embryos with mutant *Tet1/2/3* could form the primitive streak at E7.5 and reach the early somite stage with recognizable headfolds and heart at E8.5 (Fig. 2g, h). However, *Tet*-TKO embryos showed retardation at E9.5 and degeneration at E10.5 (Fig. 2i). These findings suggested that zygotic inactivation of TET enzymes leads to mid-gestation embryonic lethality, which is distinct from severe gastrulation failure of germline conditional *Tet*-TKO embryos[27]. Given that maternal TET3 is involved in active demethylation in the pronucleus of zygotes and has been removed in oocytes for producing conditional germline knock-out embryos[6,28,29], these results together indicate that the maternal TET-mediated active demethylation in zygotes is critical for embryo gastrulation. Interestingly, a recent study shows that removing TET proteins at E7.5 also leads to mid-gestation embryonic lethality[30], a phenotype similar to IMGZ-derived *Tet*-TKO embryos in our study. These data demonstrate that the IMGZ system could inactivate multiple genes in mouse embryos simultaneously without disrupting maternal factors, providing a method for unveiling the functions of multiple genes in embryogenesis.

### *Dnmt*-null embryos display gastrulation defects

We next attempted to reveal the function of DNA methylation during embryogenesis using *Dnmt*-null embryos generated by IMGZ. Pre-selection analysis identified that Dnmt1-sg1, Dnmt3a-sg3, and Dnmt3b-sg2 induced the highest efficiency of base conversion at the targeted sites, which may cause the loss of methyltransferase activity theoretically (Fig. 3a, b; Supplementary Fig. 5a and Supplementary Data 4). We first obtained *Dnmt1*-KO and *Dnmt3a/3b*-DKO embryos by injecting Dnmt1-sg1 or both Dnmt3a-sg3 and Dnmt3b-sg2 into zygotes. Western blot showed that DNMT1 or DNMT3A/3B was disrupted in these embryos (Fig. 3c). As expected, *Dnmt1* or *Dnmt3a/3b* knockout embryos significantly decreased 5mC and 5hmC signals, but demethylation of imprinting regions occurred only at *Dnmt1*-KO embryos (Fig. 3d; Supplementary Fig. 5b and Supplementary Data 3). Morphological results showed that *Dnmt1*-KO and *Dnmt3a/3b*-DKO embryos underwent normal gastrulation but were arrested at E8.5, with abnormal structure at E9.5 and almost degeneration at E10.5, which are consistent with previous studies (Fig. 3e, f and Supplementary Fig. 5c)[7,8,31].

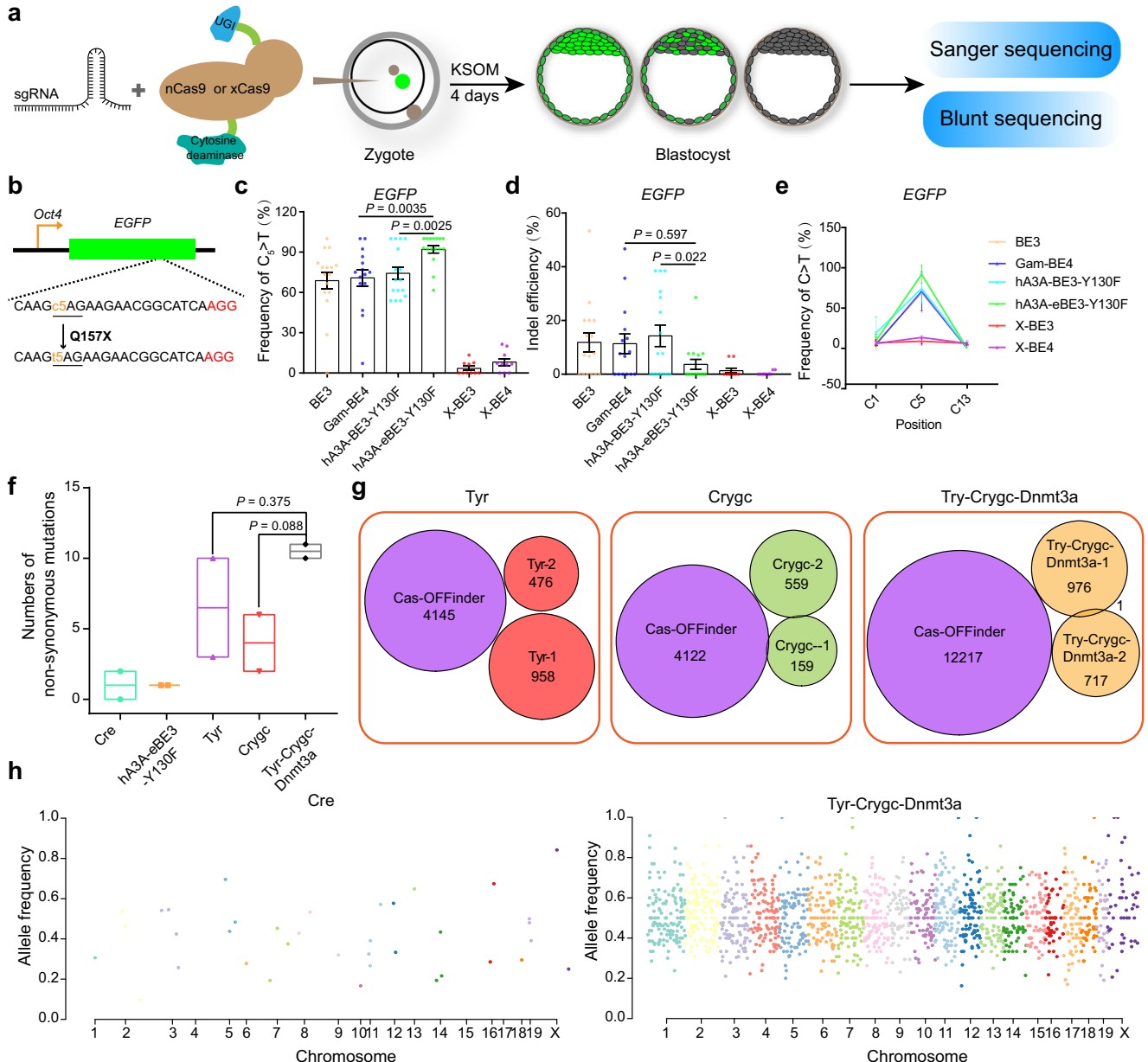

**Fig. 1 | Screening for a highly efficient base editor and analyzing its off-target effects. a** Schematic diagram of screening for highly efficient base editors that induce homozygous stop codons in early embryos through direct injection into zygotes. KSOM, embryo culture medium. **b** The target sequence in *EGFP*. The PAM sequence and the intended mutant bases are shown in red and yellow, respectively. **c, d, e** Statistical analysis of on-target C-to-T base conversion (**c**), indels (**d**) and adjacent site mutation (**e**) induced by BE3, Gam-BE4, hA3A-BE3-Y130F, hA3A-eBE3-Y130F, X-BE3, and X-BE4 in *EGFP*. Data are mean ± s.e.m for the indicated numbers of blastocysts. Detailed data are shown in Supplementary Data 1. *P* values were determined by Student's unpaired two-sided *t*-test. **f** Number of nonsynonymous mutations detected in E11.5 embryos from five groups. Floating bars show minima to maxima from two biological replicates. *P* values were determined by Student's

unpaired two-sided *t*-test; Lists of nonsynonymous off-target mutations are presented in Supplementary Data 2. **g** Analysis of SNVs detected in two E11.5 embryos with mutant *Tyr*, *Crygc*, or/and *Dnmt3a* by GOIT and off-targets predicted by Cas-OFFinder based on *Tyr*-sgRNA, *Crygc*-sgRNA, and *Dnmt3a*-sg3, indicating no overlapped sites caused by hA3A-eBE3-Y130F. The numbers present the predicted off-target sites or detected SNVs. **h** The Manhattan plot shows the distribution of all off-target mutations of different groups (Cre and Try-Crygc-Dnmt3a) in the genome. The abscissa is the location of the mutations on the chromosome, and the ordinate is the frequency of each mutation. Each point presents a mutation (SNV or indel), and the different colors indicate different chromosomes. Source data are provided as a Source Data file.

We then co-injected three sgRNAs into zygotes to produce *Dnmt*-null embryos (Fig. 3b). Consistently, injected embryos developed into blastocysts normally, and IMGZ could introduce a stop codon into *Dnmt* genes efficiently (11 of 16 analyzed blastocysts were *Dnmt*-null) (Supplementary Fig. 5d-f). After transplantation into recipient females, *Dnmt*-null embryos were morphologically indistinguishable from control embryos at E6.5, suggesting that DNMT enzymes may not be essential for egg cylinder formation (Supplementary Fig. 5d and

Supplementary Data 5). However, by E7.5, the *Dnmt*-null embryos showed obvious developmental abnormalities and primitive streak elongation failure, indicating that DNA methylation is critical for embryo gastrulation (Fig. 3e, f). LC-MS analysis indicated that both 5mC and 5hmC signals were almost zero in *Dnmt*-null embryos (Fig. 3d). These observations were confirmed by *Dnmt*-null embryos derived through mating germline-specific knockout parents (*Dnmt1^{f/+}Dnmt3a^{f/-}Dnmt3b^{f/-}; Stra8-Cre♂* x *Dnmt1^{f/-}Dnmt3a^{f/-} Dnmt3b^{f/-};*

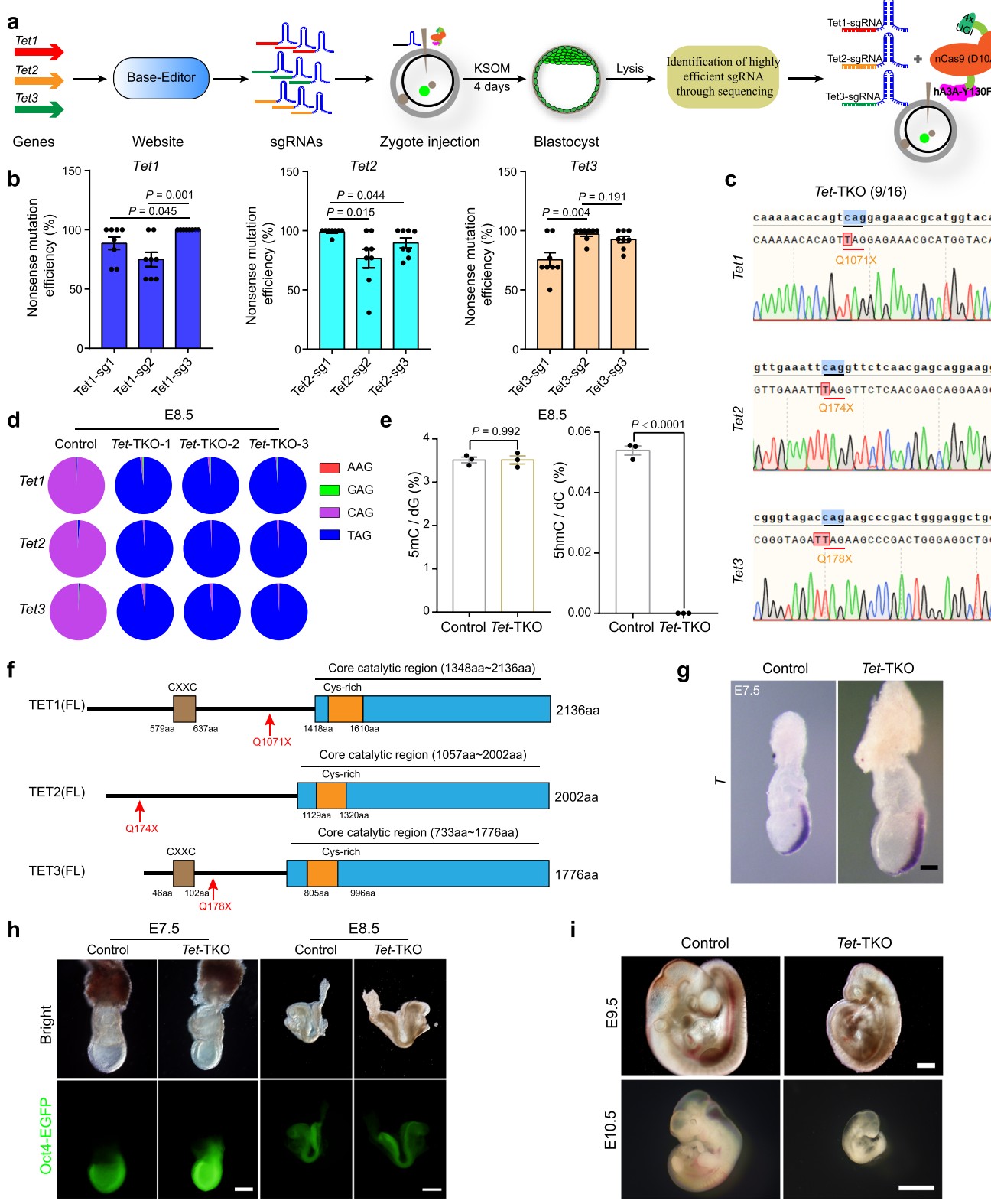

**a** Genes — Website — sgRNAs — Zygote injection — KSOM 4 days — Blastocyst — Lysis — Identification of highly efficient sgRNA through sequencing

**b** *Tet1*, *Tet2*, *Tet3* Nonsense mutation efficiency (%)

**c** *Tet*-TKO (9/16)

**d** E8.5 Control, *Tet*-TKO-1, *Tet*-TKO-2, *Tet*-TKO-3 (*Tet1*, *Tet2*, *Tet3*) — AAG, GAG, CAG, TAG

**e** E8.5

**f** TET1(FL), TET2(FL), TET3(FL)

**g** Control, *Tet*-TKO E7.5

**h** E7.5 / E8.5 Control, *Tet*-TKO — Bright / Oct4-EGFP

**i** Control, *Tet*-TKO E9.5 / E10.5

*Zp3-Cre♀*) (Fig. 3g and Supplementary Fig. 5g, h). Taken together, these data demonstrate that DNMT-mediated DNA methylation is critical for gastrulation and retention of DNMT1 or DNMT3A/3B can ensure the occurrence of gastrulation.

## Downregulation of gastrulation pathways in *Dnmt*-null embryos

We next analyzed the transcriptomic changes caused by *Dnmt* mutations by dissecting extraembryonic ectoderm (Exe) and epiblast (Epi)

of E6.5 and E7.5 embryos with different mutations (including *Dnmt1*-KO, *Dnmt3a/3b*-DKO, and *Dnmt*-null) for RNA-seq (Fig. 4a). The results showed that all Epi samples including controls and embryos with different mutants were clustered together, but separated from Exe samples which were also exclusively clustered together (Fig. 4b), indicating that the transcriptome differences between all samples are mainly caused by their different lineage origins. Expectedly, the key signal pathways that control gastrulation and primitive streak

**Fig. 2 | Establishment of the IMGZ system. a** Schematic diagram of the IMGZ system for inactivation of multiple genes in one step in embryos. **b** The editing efficiency of on-target C-to-T base conversion induced by different sgRNAs in *Tet1*, *Tet2*, and *Tet3*. Detailed data are presented in Supplementary Data 4. Data are mean ± s.e.m for the indicated numbers of blastocysts. *P* values were determined by Student's unpaired two-sided *t*-test. **c** Representative Sanger sequence chromatograms of one *Tet1/2/3*-TKO blastocyst. 9 of 16 checked blastocysts show homozygous mutant chromatograms. **d** Deep sequencing analysis of genome mutant sites of one control and three *Tet1/2/3*-TKO embryos (E8.5). The results indicated the conversion of CAG codon of *Tet1/2/3* into TAG stop codon in resultant embryos. **e** Frequencies of 5mC and 5hmC modified nucleotides in the genomic DNA of control and *Tet1/2/3*-TKO embryos (E8.5) were determined by quantitative mass spectrometry. Data are mean ± s.e.m of three biological replicates. *P* values were determined by Student's unpaired two-sided *t*-test. **f** Domain structure of

full-length TET1, TET2, and TET3 proteins. The red arrows indicate amino acid mutation sites produced by the sgRNAs with the highest base conversion efficiency. **g** RNA in situ hybridization of *T* (also known as brachyury, a primitive streak marker) in the control and *Tet*-TKO embryos (E7.5). Three independent embryos were analyzed for each group. Scale bar, 100 μm. **h** Representative images of IMGZ-derived control and *Tet*-TKO embryos at E7.5 and E8.5. Green fluorescence indicates the expression of *Oct4-EGFP*. Three independent embryos were analyzed for each group. Scale bar, 200 μm in the left panel and 500 μm in the right panel. **i** Representative images of control and *Tet*-TKO embryos at E9.5 and E10.5 generated by the IMGZ system. Green fluorescence indicates the expression of *Oct4-EGFP*. Three independent embryos were analyzed for each group. Scale bars, 500 μm in the up panel and 2 mm in the down panel. Source data are provided as a Source Data file.

formation in mice[32–35] were misregulated in *Dnmt*-null embryos (Supplementary Fig. 6a). Meanwhile, the marker genes involved in the primitive streak formation and germ layer determination were also aberrant in *Dnmt*-null embryos (Supplementary Fig. 6b). Moreover, the global germ-layer enhancer-related genes for ectoderm, mesoderm, and definitive endoderm[36] showed misregulation in *Dnmt*-null Epi compared to both *Dnmt1*-KO and *Dnmt3a/3b*-DKO samples at E7.5 (Supplementary Fig. 7a-c).

We next investigated the common differentially expressed genes (DEGs) in *Dnmt*-null embryos compared to both *Dnmt1*-KO and *Dnmt3a/3b*-DKO and found that pathways mainly involved in post-implantation development and metabolism were disrupted in *Dnmt*-null embryos (Fig. 4c–f). Surprisingly, the affected pathways related to gastrulation were down-regulated in *Dnmt*-null embryos (Fig. 4c–f). Interestingly, we observed that germline-related genes were further de-repressed in E7.5 *Dnmt*-null Epi compared to *Dnmt1*-KO and *Dnmt3a/3b*-DKO embryos (Fig. 4f), which is consistent with recent findings which report that the silencing of these genes during post-implantation stages are mainly regulated by DNA methylation[31,37]. In contrast to our knowledge that DNA demethylation may cause gene up-regulation, gastrulation-related pathways are down-regulated in *Dnmt*-null embryos, suggesting that other factors associated with DNA methylation are involved in this suppression.

## DNA methylation underlies gastrulation

Given that oxidation of 5-methylcytosine by the TET family of dioxygenases can lead to demethylation[4,5], it is interesting to investigate the functional significance of both cytosine methylation and demethylation in mouse embryogenesis by producing *Tet/Dnmt*−6KO embryos through IMGZ. To achieve this, we first obtained methylation/demethylation free (*Tet/Dnmt*−6KO) embryos through co-injection of six sgRNAs targeting *Dnmt1/3a/3b* and *Tet1/2/3* with the highest mutation efficiency respectively into zygotes (Supplementary Fig. 8a). The results showed that *Tet/Dnmt*−6KO embryos exhibited normal pre-implantation development (4 of 24 analyzed blastocysts were *Tet/Dnmt*−6KO) (Supplementary Data 6), formed normal egg cylinders at E6.5, and displayed severely developmental defects at E7.5 (Fig. 5a; Supplementary Fig. 8b, c and Supplementary Data 7). Besides, we found that the areas of epiblast and the primitive streak of *Dnmt*-null and *Tet/Dnmt*−6KO were similar, but both were smaller than control embryos at E7.5 (Supplementary Fig. 8d-f). These results showed that *Dnmt*-null and *Tet/Dnmt*−6KO embryos are overall similar, suggesting that TET's functions during embryonic development are dependent on DNA methylation.

Interestingly, when *Dnmt1* or *Dnmt3a/3b* are intact, *Tet/Dnmt3a/3b*−5KO or *Tet/Dnmt1*−4KO embryos exhibited normal gastrulation but failed to develop through E8.5, the phenotypes similar to those observed in corresponding *Dnmt3a/3b*-DKO and *Dnmt1*-KO embryos (Fig. 5a, b; Supplementary Fig. 8a, g and Supplementary Data 7). Moreover, LC-MS analysis indicated that while 5hmC signals vanished

in both *Tet/Dnmt3a/3b*−5KO and *Tet/Dnmt1*−4KO embryos, 5mC signals in the *Tet/Dnmt1*-4KO and *Tet/Dnmt3a/3b*-5KO embryos appeared similar to those in the *Dnmt1*-KO and *Dnmt3a/3b*-DKO (0.51% *vs* 0.49% and 1.04% *vs* 1.10%), respectively, suggesting that the DNA methylation protected by DNMT1 or DNMT3A/3B is independent of TET proteins (Figs. 3d, 5c and Supplementary Data 3). Further, whole genome bisulfite sequencing (WGBS) analysis showed that the resultant E7.5 embryos with the same *Dnmt* mutations with/without *Tet*-TKO were tightly clustered together (Fig. 5d). The average CG methylation level dropped from 77.8% in controls to around 20% in single or double *Dnmt* mutants with/without *Tet*-TKO (Fig. 5e and Supplementary Data 8). These results suggest that there are methylated sites in the genome maintained by DNMT1 or DNMT3A/3B that may be involved in the embryo gastrulation, which is not associated with DNA demethylation mediated by TET proteins (Fig. 5f).

## miRNA promoter methylation may be related to gastrulation

We next sought to investigate the common genomic sites methylated either by DNMT1 or DNMT3A/3B that may underlie the down-regulation of gastrulation-related pathways in *Dnmt*-null embryos. WGBS analysis of Epi and Exe showed dramatically decreased methylation levels in both Epi and Exe of *Dnmt1* or *Dnmt3a/3b* while almost zero in *Dnmt*-null embryos (Fig. 6a and Supplementary Data 8). Meanwhile, samples with the same mutant type tightly clustered together (Supplementary Fig. 9a). Interestingly, E6.5 and E7.5 *Dnmt3a/3b*-DKO samples were much closer to control blastocysts, suggesting that the DNMT1 can sustainably maintain DNA methylation remained in blastocysts to the late-stage embryos (Supplementary Fig. 9a)[38]. Meanwhile, E6.5 and E7.5 *Dnmt1*-KO samples were much closer to E5.5 and E6.5 control samples, suggesting that de novo methylation mediated by DNMT3A/3B plays critical roles after the blastocyst stage, consistent with previous observations that DNMT3A/3B are responsible for methylation acquisition in embryonic development[7,31].

We next analyzed the methylation changes in all genomic components, including promoters, enhancers, exons, introns, intragenic regions, intergenic regions, CpG islands (CGIs), and transposable elements (TEs). As expected, inactivation of *Dnmt1* or *Dnmt3a/3b* induced global reduction of DNA methylation across all genomic sequences in *Dnmt1*-KO or *Dnmt3a/3b*-DKO embryos, respectively (Supplementary Fig. 9b). Given that the methylation levels of TEs, enhancers, and promoters can affect the expression of related elements or genes during early embryonic development[31,36,39], we thus investigated the patterns of overlapped hypermethylated elements between *Dnmt1*-KO and *Dnmt3a/3b*-DKO embryos compared to *Dnmt*-null embryos (Fig. 6b). The results showed obviously increased DNA methylation levels of these overlapped elements in *Dnmt1*-KO compared to those in wild-type blastocysts (Supplementary Fig. 10a-c). Interestingly, we also found that *Dnmt3a/3b*-DKO displayed significantly increased levels of DNA methylation compared to blastocysts (Supplementary Fig. 10a-c),

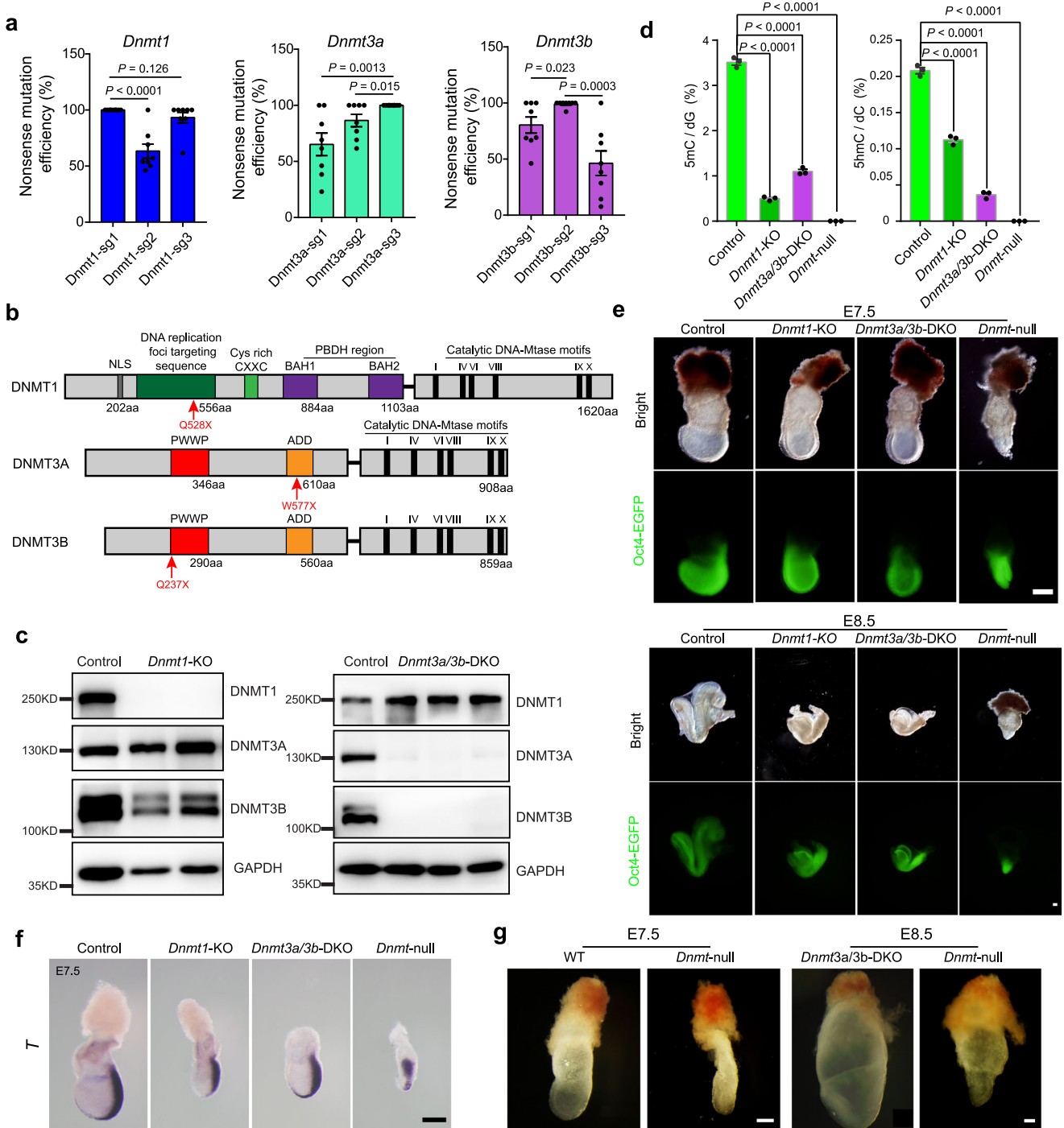

**Fig. 3 | One-step inactivation of *Dnmt1/3a/3b* mediated by IMGZ reveals critical roles of DNA methylation in mouse gastrulation. a** The editing efficiencies of on-target C-to-T base conversion induced by different sgRNAs in *Dnmt1*, *Dnmt3a*, and *Dnmt3b*. Detailed data are presented in Supplementary Data 4. Data are mean ± s.e.m for the indicated numbers of blastocysts. *P* values were determined by Student's unpaired two-sided *t*-test. **b** Domain structure of full-length DNMT1, DNMT3A, and DNMT3B proteins. The red arrows indicate the amino acid mutation site produced by the sgRNAs with the highest base conversion efficiency. **c** Expression of DNMT1, DNMT3A, and DNMT3B in *Dnmt1*-KO and *Dnmt3a/3b*-DKO embryos. GAPDH is the control. Proteins were obtained from E9.5 embryo lysis. Two control, two *Dnmt1*-KO, and three *Dnmt3a/3b*-DKO embryos were analyzed. **d** Frequencies of 5mC and 5hmC modified nucleotides in the genomic DNA of control, *Dnmt1*-KO, *Dnmt3a/3b*-DKO, and *Dnmt*-null embryos (E8.5) were

determined by quantitative mass spectrometry. Data are mean ± s.e.m of three biological replicates. *P*-values were determined by Student's unpaired two-sided *t*-test. **e** Representative images of IMGZ-derived control, *Dnmt1*-KO, *Dnmt3a/3b*-DKO, and *Dnmt*-null embryos at E7.5 and E8.5, showing developmental retardation at E7.5 in *Dnmt*-null embryos. Green fluorescence indicates the expression of *Oct4-EGFP*. Three independent embryos were analyzed for each group. Scale bars, 200 μm. **f** RNA in situ hybridization of *T* in the control, *Dnmt1*-KO, *Dnmt3a/3b*-DKO, and *Dnmt*-null embryos (E7.5), showing primitive streak elongation failure in *Dnmt*-null embryos. Three independent embryos were analyzed for each group. Scale bar, 100 μm. **g** Representative images of E7.5 and E8.5 *Dnmt*-null embryos obtained through crossing germline-specific conditional knockout parents. Scale bars, 200 μm. Three independent embryos were analyzed for each group. Source data are provided as a Source Data file.

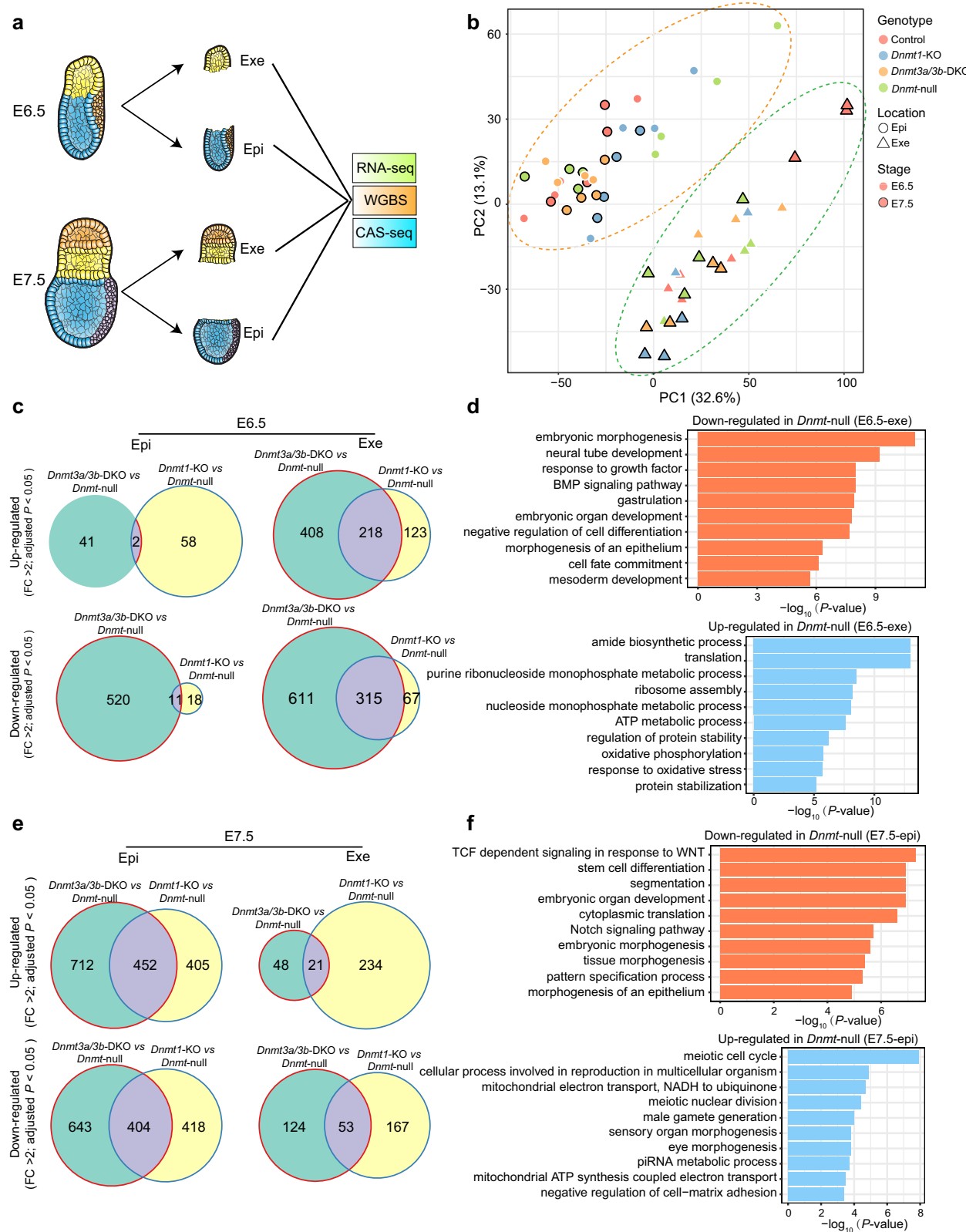

suggesting that DNMT1 may have de novo methylation activity at these sites as shown in recent reports[40,41], or probably DNMT1 may function more efficiently at maintaining inherited DNA methylation during the absence of DNMT3A/3B proteins. Meanwhile, these common hypermethylated elements mediated by DNMT1 or DNMT3A/3B were mainly distributed at Epi (Fig. 6b), implying critical roles of DNA methylation in Epi development. This may also reflect the differences in DNA

methylation patterns between Epi and Exe lineages, in which partially methylated domains (PMDs) span the majority of the genome while the highly methylated genomes in somatic cells[42]. Interestingly, a recent report indicates that de novo DNA methylation mediated by DNMT3B plays an essential role in placental development and function[43], probably through depositing DNA methylation at these PMDs.

**Fig. 4 | Transcriptome analysis of *Dnmt* mutant embryos. a** Schematic diagram of collecting Exe and Epi at E6.5 or E7.5 for RNA sequencing (RNA-seq), whole genome bisulfite sequencing (WGBS), and Cas9-assisted small RNA-sequencing (CAS-seq). Exe, extraembryonic ectoderm; Epi, epiblast. **b** Principal component analysis (PCA) of RNA-seq data from E6.5 and E7.5 embryos with different mutations. **c** Venn diagrams show up-regulated and down-regulated differentially expressed genes (DEGs) in *Dnmt1*-KO and *Dnmt3a/3b*-DKO compared to *Dnmt*-null of Exe and Epi at E6.5. *P*-values were calculated by Deseq2. **d** Enriched Gene Ontology terms of overlapped down-regulated (up) and up-regulated (down) DEGs (*Dnmt*-null *vs Dnmt1*-KO and *Dnmt*-null *vs Dnmt3a/3b*-DKO) in E6.5 *Dnmt*-null Exe related to (**c**). *P* values were calculated by Metascape. **e** Venn diagrams show up-regulated and down-regulated DEGs in *Dnmt1*-KO and *Dnmt3a/3b*-DKO compared to *Dnmt*-null of Exe and Epi at E7.5. *P* values were calculated by Deseq2. **f** Enriched Gene Ontology terms of overlapped down-regulated (up) and up-regulated (down) DEGs (*Dnmt*-null *vs Dnmt1*-KO and *Dnmt*-null *vs Dnmt3a/3b*-DKO) in E7.5 *Dnmt*-null Epi related to (**e**). *P* values were calculated by Metascape. Source data are provided as a Source Data file.

The majority of overlapped hypermethylated TEs were retrotransposons, including LTR, SINE, and LINE families (Supplementary Fig. 11a). RNA-seq analysis indicated that while LTRs were dramatically up-regulated in all mutant samples, which further altered the expression of proximal genes (Supplementary Fig. 11b, c), the common up-regulated LTRs mediated by DNA methylation in *Dnmt*-null compared to *Dnmt1*-KO and *Dnmt3a/3b*-DKO embryos slightly influenced proximal genes (Supplementary Fig. 12a-c). Moreover, the affected genes by the overlapped hypermethylated enhancers (HMEs) were mainly related to Epi samples and enriched in organ and tissue development (Supplementary Fig. 12d-f). Taken together, these results suggest that DNA methylation of retrotransposons and enhancers may not be critical for gastrulation.

Given that during peri-implantation DNA methylation in the epiblast is targeted to the promoter of lineage-specific genes[44], we further characterized the methylation patterns of promoter regions (TSS ± 2kb) in mutant embryos (Fig. 6c). The results showed that both DNMT1 and DNMT3A/3B contributed to DNA methylation at promoters in E6.5/E7.5 embryos (Fig. 6b, c). Interestingly, the overlapped hypermethylated promoters (HMPs) sustained significantly higher DNA methylation compared to non-HMPs in controls, and DNMT1 or DNMT3A/3B could maintain hypermethylation at these loci in *Dnmt1*-KO or *Dnmt3a/3b*-DKO embryos respectively (Fig. 6d). Around 28% of the overlapped HMP-related genes were protein-coding genes; however, there was no obvious correlation between methylation state and gene expression (Fig. 6e and Supplementary Fig. 13a, b). Strikingly, the majority of genes related to these overlapped HMPs were short and lacked CGI in their promoters, including miRNAs (46.7%) and lncRNA (around 18.9%) (Fig. 6e and Supplementary Fig. 13c, d).

### Overexpression of miRNAs is related to gastrulation failure

We next investigated the profiling of miRNA expression by performing Cas9-assisted small RNA-sequencing (CAS-seq)[45] on E6.5 and E7.5 of Exe and Epi samples with different mutations (Fig. 4a). The results showed that all samples fell into four groups according to their developmental origins and stages (Supplementary Fig. 14a), suggesting that DNA methylation overall did not affect the miRNA profiling of early embryos. However, DEG analysis showed many significantly upregulated and downregulated miRNAs in *Dnmt*-null embryos compared to other groups (Supplementary Fig. 14b). Strikingly, the majority of HMP-related DEGs were significantly up-regulated in *Dnmt*-null embryos (Fig. 7a and Supplementary Fig. 14c, d). Besides, RNA-seq data revealed no significant differences in miRNA biogenesis pathways and RNA binding proteins among samples with different mutants (Supplementary Fig. 15a-c). Taken together, these observations suggest that DNA methylation may directly influence miRNA expression.

A total of 54 HMP-related up-regulated miRNAs were identified in *Dnmt*-null embryos compared to controls, including *let-7b-5p* and *miR-296-3p* (Supplementary Fig. 14c), which have shown their roles in germ layer specification and embryonic stem cell differentiation[46,47]. Interestingly, the remaining 52 miRNAs were all located at the genomic imprinting *Dlk1-Dio3* region (Supplementary Fig. 16a). Next, we used TargetScan to predict targets of these up-regulated miRNAs. Strikingly, large proportions of the down-regulated genes in *Dnmt*-null embryos compared to control samples revealed by RNA-seq data were computationally predicted targets of these miRNAs, especially in Epi samples (Fig. 7b). Interestingly, these predicted targets were enriched in embryogenesis and metabolism pathways, including glucose pathways that are involved in gastrulation[48] in E6.5 and E7.5 Epi samples (Fig. 7c). Our data suggest that the up-regulated miRNAs caused by the removal of DNA methylation may partially influence gastrulation through inhibiting related pathways.

### Suppression of miRNAs fine-tunes gastrulation

*let-7b-5p* was significantly upregulated in *Dnmt*-null samples compared to the control, *Dnmt1*-KO, or *Dnmt3a/3b*-DKO samples (Supplementary Fig. 14c). *let-7b-5p* plays a critical role in germ layer specification[46]. Overexpression of *let-7b-5p* promotes stem cell differentiation, while in cancer cells, it is suppressed, and its activation can inhibit cancer cell proliferation, suggesting that precise miRNA dosage might be critical for physiological and pathological processes[49]. WGBS analysis showed that promoters of *let-7b-5p* and *miR-296-3p* retained hypermethylation in both *Dnmt1*-KO and *Dnmt3a/3b*-DKO embryos (Supplementary Fig. 16b), implying that promoter DNA methylation may be involved in suppression of miRNA expression.

More than 60 miRNAs have been identified in the murine *Dlk1-Dio3* locus[50], which are maternally expressed and involved in the development and disease[50–53]. We next compared the expression of *Dlk1-Dio3* miRNA in all samples and found that the majority of miRNAs were significantly upregulated in *Dnmt*-null embryos when compared to control and *Dnmt3a/3b*-DKO, while less when compared to *Dnmt1*-KO embryos (Supplementary Fig. 16c). WGBS analysis showed that promoters of these miRNAs retained hypermethylation in both *Dnmt1*-KO and *Dnmt3a/3b*-DKO embryos (Supplementary Fig. 17a, b). Meanwhile, DNA methylation of intergenic germline-derived (*IG*)-DMR, especially the IG[CGI] region[54], was sustained in *Dnmt3a/3b*-DKO embryos but was lost in *Dnmt1*-KO and *Dnmt*-null embryos (Fig. 7d), indicating that DNMT1 is involved in DNA methylation maintenance of germline DMRs. In summary, these observations suggest that *IG*-DMR methylation is not related to promoter methylations that are mediated by either DNMT1 or DNMT3A/3B. However, suppression of miRNA expression may be mainly related to *IG*-DMR methylation in the *Dlk1-Dio3* locus. Moreover, promoter methylations could also be partially involved in the suppression of miRNAs. Consistent with our observations, in mature oocytes, where DNA methylation was partially maintained at *IG*-DMR but with high levels at promoters, the majority of miRNAs at *Dlk1-Dio3* were not expressed (Supplementary Fig. 18a-d)[55,56]. Meanwhile, preimplantation embryos kept the hypermethylation state at these promoters[55] (Supplementary Fig. 18a), and they sustained a similar suppression state of *Dlk1-Dio3* miRNAs as oocytes[56] (Supplementary Fig. 18c, d). These observations indicate that dramatic demethylation after fertilization doesn't change the epigenetic state of *Dlk1-Dio3* miRNAs, suggesting that maintenance of promoter methylation and suppression of *Dlk1-Dio3* miRNAs may be critical for early embryonic development.

To determine whether the dosage of miRNAs directly contributes to gastrulation failure, we adopted androgenetic haploid embryonic stem cells (AG-haESCs) that carry sperm genome and can be used as sperm replacement to support embryonic development[57]. AG-haESCs were cultured under conditions with 2i (inhibitors of Mek1/2 and

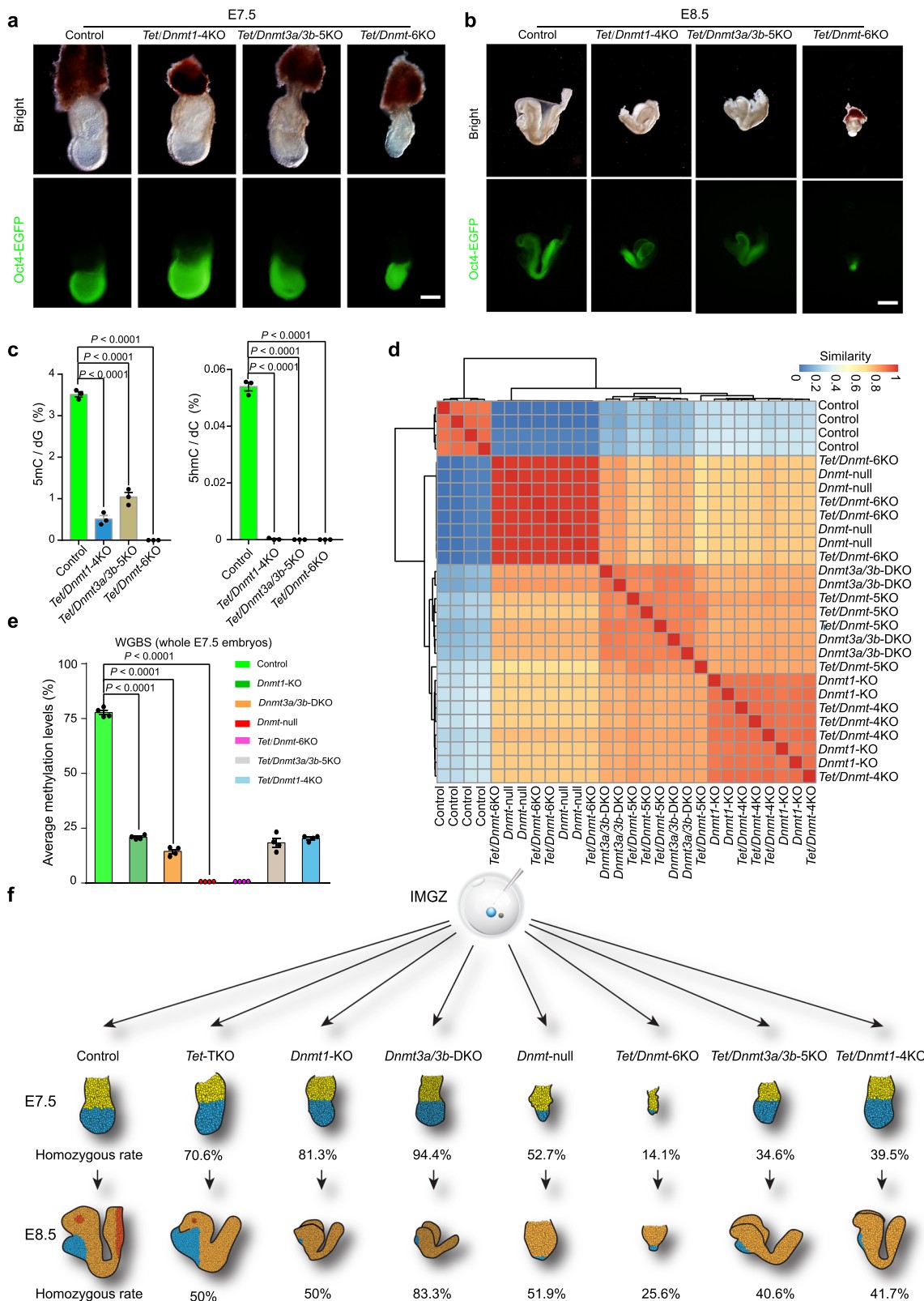

Gsk3β), which induce loss of paternal DMR methylations, including *H19* and *IG* DMRs, and impair developmental potential in AG-haESCs[58,59]. After the removal of both *H19* and *IG* DMRs to partially mimic the DMR methylation state, the resulting AG-haESCs (termed DKO-AG-haESCs) can efficiently support full-term embryonic development after injection into oocytes[58–60]. We performed Cas-seq of small RNAs on wild-type and DKO-AG-haESCs and found that while

*IG*-DMR deletion didn't increase the methylation level of the promoters (Supplementary Fig. 18e)[58,59], this deletion induced significantly reduced expression of *Dlk1-Dio3* miRNAs in DKO-AG-haESCs (Fig. 7e and Supplementary Fig. 18f, g). Besides, after injection of AG-haESCs into oocytes, the resulting semi-cloned (SC) embryos from DKO-AG-haESCs showed suppression of *Dlk1-Dio3* miRNA expression compared to SC embryos from wild-type cells with hypomethylated *IG*-DMR

**Fig. 5 | DNA methylation is involved in gastrulation independent of DNA demethylation. a, b** Representative images of IMGZ-derived E7.5 (**a**) and E8.5 (**b**) embryos carrying *Tet1/2/3;Dnmt1*−4KO, *Tet1/2/3;Dnmt3a/3b*−5KO, or *Tet1/2/3;Dnmt1/3a/3b*−6KO. Green fluorescence indicates the expression of *Oct4-EGFP*. Three independent embryos were analyzed for each group. Scale bars, 200 μm in (**a**) and 500 μm in (**b**). **c** Frequencies of 5mC and 5hmC modified nucleotides in the genomic DNA of control, *Tet1/2/3;Dnmt1*−4KO, *Tet1/2/3;Dnmt3a/3b*−5KO, and *Tet1/2/3;Dnmt1/3a/3b*−6KO embryos (E8.5) were determined by quantitative mass spectrometry. Data are mean ± s.e.m of three biological replicates. *P*-values were

determined by Student's unpaired two-sided *t*-test. **d** Heatmap shows the similarity of WGBS data between different mutant embryos. The similarity between samples was defined as min-max normalized Euclidean distance reduced by 1. **e** Bar plots represent the average CG methylation level of whole embryos (E7.5) with different mutations measured by WGBS. Data are mean ± s.e.m of four biological replicates. *P*-values were determined by Student's unpaired two-sided *t*-test. **f** Summary of phenotypes observed in E7.5 and E8.5 embryos carrying different mutant *Dnmt* and *Tet* genes generated by IMGZ. Detailed data of homozygous mutants are presented in Supplementary Data 6. Source data are provided as a Source Data file.

(Fig. 7f), indicating that *IG*-DMR methylation state indeed suppresses *Dlk1-Dio3* miRNA expression. Nevertheless, after removing all DNA methylation, including promoter methylation through IMGZ in SC embryos derived from wild-type AG-haESCs, we found obviously an increased expression of *Dlk1-Dio3* miRNAs (Fig. 7g and Supplementary Fig. 18h). Besides, SC embryos from DKO-AG-haESCs after removal of *Dnmts* exhibited increased *let-7b-5p* and *Dlk1-Dio3* miRNAs compared to DKO-AG-haESCs derived SC embryos (Supplementary Fig. 18i, j). Given that hypomethylation at *IG*-DMR in AG-haESCs can't be re-established in resultant SC embryos[58], these results together imply that promoter methylation can be established by DNMT1 and/or DNMT3A/3B during early embryonic development and may also be partially involved in inhibition of *Dlk1-Dio3* miRNAs. Consistently, the *Dnmt*-null SC embryos from DKO-AG-haESCs also showed gastrulation failure at E7.5 (Fig. 7h and Supplementary Data 6), indicating that *IG*-DMR deletion can't rescue gastrulation phenotypes in *Dnmt*-null SC embryos. Finally, we deleted six commonly up-regulated miRNAs identified in *Dnmt*-null embryos when compared to *Dnmt1*-KO, *Dnmt3a/3b*-DKO, and control embryos, including *let-7b* and 5 *Dlk1-Dio3* miRNAs (*miR-127*, *miR-541*, *miR-369*, *miR-540*, and *miR-409*) (Supplementary Fig. 14c, d), in DKO-AG-haESCs (Supplementary Fig. 19a-c). Strikingly, these heterozygous deletions almost completely suppressed the expression of *Dlk1-Dio3* miRNAs in *Dnmt*-null SC embryos (Fig. 7i and Supplementary Fig. 19d). As a consequence, the morphogenesis and elongation of the primitive streak were partially rescued at E7.5, although the development of these embryos still failed at E8.5 (Fig. 7h, j and Supplementary Data 6). Taken together, these data suggest the role of DNA methylation of promoters in suppressing the expression of miRNAs, which may fine-tune the expression of transcriptional factors or gastrulation-related genes to safeguard embryo development (Supplementary Fig. 19e).

## Discussion

During embryogenesis, complex regulators, including epigenetic and transcriptional factors, cooperatively regulate lineage differentiation, tissue patterning, and organogenesis[61–63]. Genetic disruption of these factors helps us understand the molecular functions of these regulators during embryogenesis[63]. However, the family gene-involved compensation mechanisms may make the findings from a single mutation difficult to interpret. Moreover, the fundamental role of a biological event, such as DNA demethylation induced by *Tet* family genes[27], cannot be revealed from mouse models with mutations in part of family genes. Therefore, it is necessary to inactivate all family genes simultaneously in zygotes to reveal the physiological roles of a biological process. The conventional strategy is to establish a complicated breeding system based on germline-specific knockout mouse models. However, these strategies are time-consuming and labor-intensive. Together, it is still an unmet goal to efficiently produce embryos with mutations in family genes. In this study, we establish a base-editor (hA3A-eBE-Y130F) mediated IMGZ system that enables the efficient generation of embryos with multiple mutant genes in one step, which can accelerate functionally deciphering the significance of a biological process during embryogenesis.

DNMTs and TETs elaborately regulate DNA methylation and demethylation in embryogenesis[4,7,8,27]. *Dnmt1* or *Dnmt3a/3b* mutant mouse models have been generated and shown their roles after gastrulation by protecting DNA methylation to repress a panel of genes and multiple retrotransposons for transcription integrity[6,7,31]. However, it is still unknown what is the function of DNA methylation in embryogenesis. Moreover, although a previous study demonstrates that *Tet* and *Dnmt* act antagonistically to regulate early development[27], it is still largely unknown the relationship between DNMT-mediate DNA methylation and TET-mediated demethylation during embryogenesis. Here, we applied the IMGZ system to generate *Tet*-TKO embryos and found that those embryos display normal gastrulation. Importantly, IMGZ-derived *Dnmt*-null embryos fail to gastrulate while *Dnmt1*-KO and *Dnmt3a/3b*-DKO embryos display normal gastrulation. These phenotypes are further confirmed by *Dnmt*-null embryos generated by the conventional germline-specific knockout strategy. Moreover, *Tet/Dnmt3a/3b*−5KO, *Tet/Dnmt1*−4KO, and *Tet/Dnmt*−6KO embryos exhibit similar phenotypes shown in embryos with only respective *Dnmt* mutations. Taken together, these observations indicate that DNMT1 or DNMT3A/3B-mediated DNA methylation controls gastrulation and this function is independent of TET-mediated DNA demethylation.

Thanks to the efficiency and reliability of mutant embryos generation by IMGZ, our system can produce sufficient numbers of samples for high-throughput analyses in one generation. By mapping genome-wide profiles of transcription, DNA methylation, and miRNA expression in *Dnmt* mutants, we uncovered a critical role of promoter DNA methylation in the suppression of miRNA expression in gastrulation, the majority of which are located in the *Dlk1-Dio3* locus. We found that DNA methylation of both *IG*-DMR and promoters contributes to the suppression of miRNA expression in the *Dlk1-Dio3* locus. Previous studies have shown that *IG*-DMR is involved in regulating the expression of the *Dlk1-Dio3* miRNA cluster, and removal of methylation at *IG*-DMR can increase miRNA expression in *Dnmt1*-KO or parthenogenesis embryos[64]. Consistently, our results also indicate the role of *IG*-DMR methylation in the repression of miRNA expression, including less up-regulated miRNA in *Dnmt*-null embryos when compared to *Dnmt1*-KO embryos than compared to *Dnmt3a/3b*-DKO embryos (Supplementary Fig. 16c) and significantly reduced expression of *Dlk1-Dio3* miRNAs in DKO-AG-haESCs induced by *IG*-DMR deletion (Fig. 7e and Supplementary Fig. 18g). However, *Dnmt1*-KO or parthenogenesis embryos die at mid-gestation[8,65], inconsistent with more severe phenotypes of gastrulation failure observed in the *Dnmt*-null embryos, suggesting that genome-wide methylation-free can influence many genetic and epigenetic events during embryogenesis, including other epigenetic modifications and even genome instability (Supplementary Fig. 19e)[38,66].

Nevertheless, in the current study, we provided multiple pieces of evidence to support a possible link between promoter methylation and miRNA expression, including the *Dlk1-Dio3* miRNA cluster, which is also involved in gastrulation. First, *IG*-DMR methylation doesn't control promoter methylation because the promoters sustain hypermethylation in both *Dnmt1*-KO and *Dnmt3a/3b*-DKO embryos (Supplementary Fig. 17a, b) and *IG*-DMR deletion in AG-haESCs doesn't increase the methylation level of the promoters (Supplementary

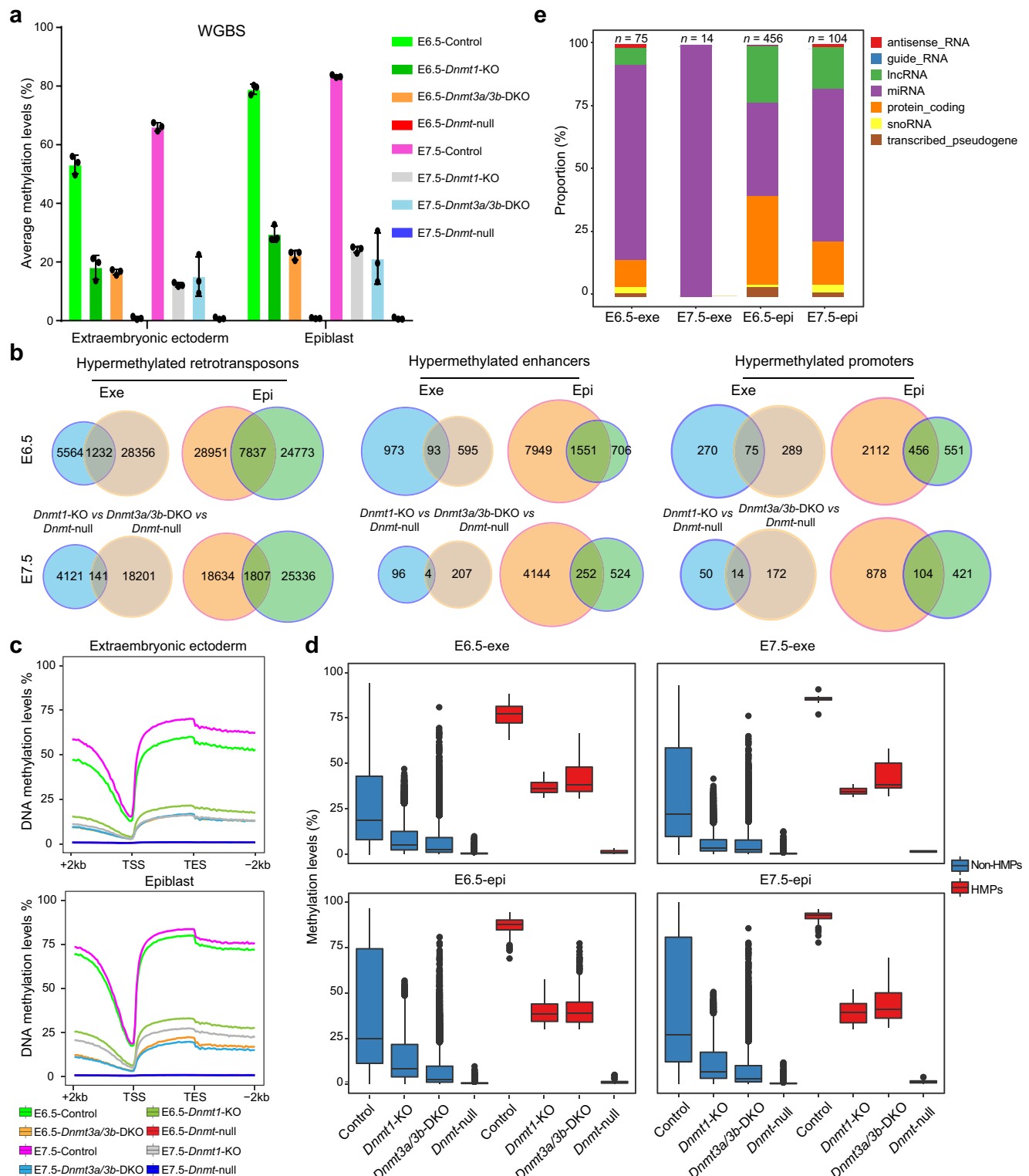

**Fig. 6 | Overlapped hypermethylated promoters mediated by DNMT1 and/or DNMT3A/3B are related to embryo gastrulation. a** Bar plots represent the average CG methylation level of Exe and Epi measured by WGBS in four groups (control, *Dnmt1*-KO, *Dnmt3a/3b*-DKO, and *Dnmt*-null embryos). Data are mean ± s.e.m of three biological replicates. **b** Venn diagrams show overlapped hypermethylated retrotransposons, enhancers, and promoters between *Dnmt1*-KO *vs Dnmt*-null and *Dnmt3a/3b*-DKO *vs Dnmt*-null in Exe and Epi of E6.5 and E7.5. **c** Average DNA methylation levels in the gene body and an additional 2kb flanking regions in four genotypes of Exe and Epi at E6.5 and E7.5. TSS, transcription start site; TES, transcription end site. **d** Box plots show promoter methylation levels of overlapped hypermethylated promoters (HMPs) and non-HMPs in embryos with different mutants. HMPs are overlapped HMPs shown in (**b**); Non-HMPs are other promoters besides the HMPs. The central lines are the median of data. The lower and upper hinges correspond to the 25th and 75th percentiles. The end of the lower and upper whiskers are 1.5 * IQR (inter-quartile range). Data beyond the end of the whiskers are plotted as outliers. **e** Bar plots show the composition of genes related to overlapped HMPs in Exe and Epi at E6.5 and E7.5 shown in (**b**). *n*, the number of overlapped HMPs. Source data are provided as a Source Data file.

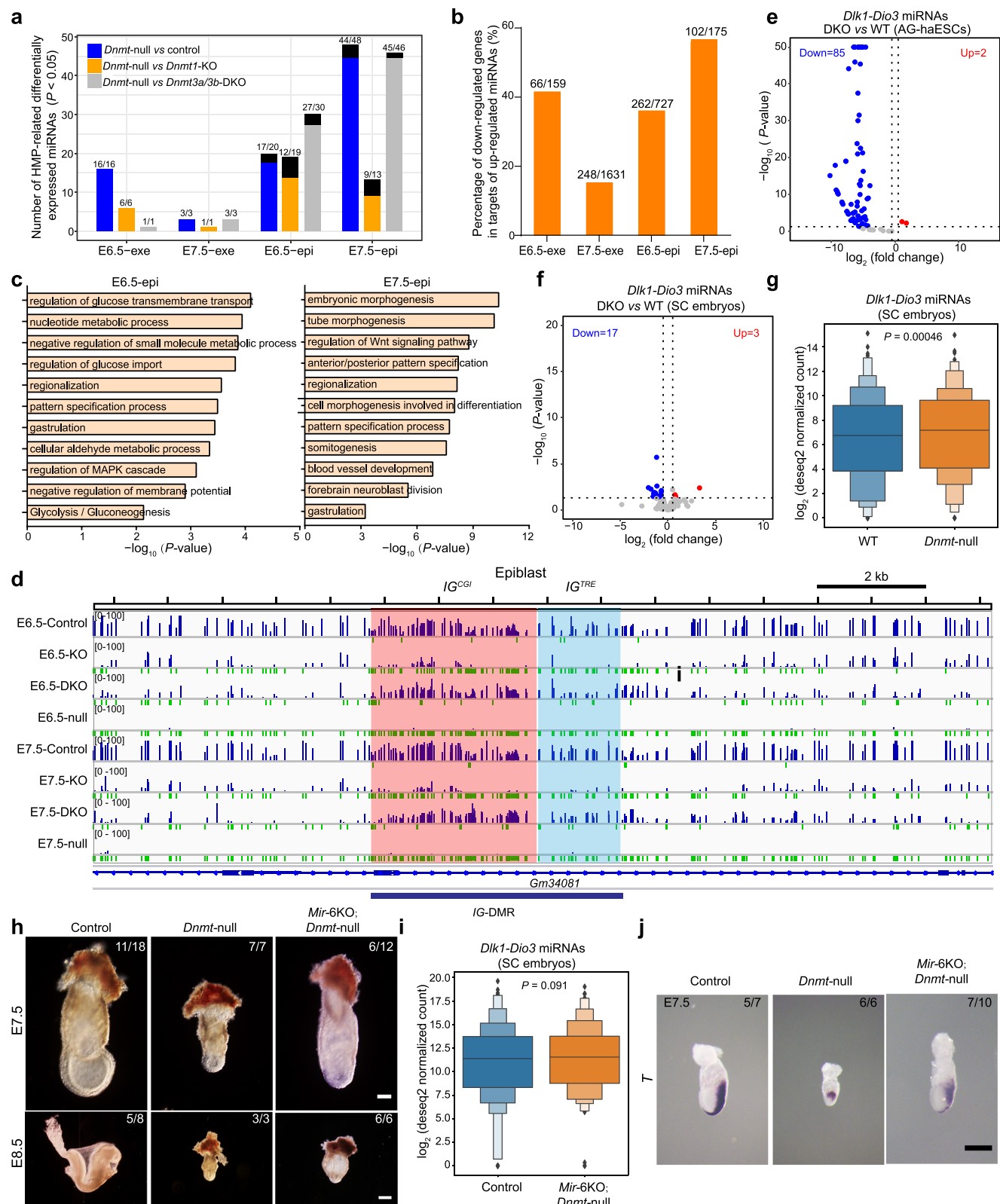

Fig. 18e). Second, in mature oocytes, where DNA methylation is partially maintained at *IG*-DMR but with high levels at promoters, the majority of miRNAs at *Dlk1-Dio3* are not expressed (Supplementary Fig. 18a-d). Third, pre-implantation embryos keep the hypermethylation state at these promoters and sustain a similar suppression state of *Dlk1-Dio3* miRNAs as oocytes (Supplementary Fig. 18c, d). Fourth, *IG*-DMR deletion that mimics the *IG*-DMR methylation can't rescue gastrulation phenotypes in *Dnmt*-null SC embryos (Fig. 7h), suggesting

that promoter methylation may also be involved in gastrulation failure. Fifth, two other HMP-related up-regulated miRNAs, *let-7b-5p* and *miR-296-3p*, are up-regulated in all tested *Dnmt*-null embryos compared to control embryos (Supplementary Fig. 14c, 16b). Finally, decreasing six commonly up-regulated miRNAs identified in *Dnmt*-null embryos, including 5 *Dlk1-Dio3* miRNAs, could partially rescue the elongation failure of the primitive streak in *Dnmt*-null embryos (Fig. 7j), indicating that the dosage of miRNAs is critical for gastrulation. *Dicer* deletion in

**Fig. 7 | Suppression of miRNA expression mediated by DNA methylation fine-tunes gastrulation. a** Bar plots show the numbers of differently expressed miRNAs related to overlapped HMPs. Numerators/denominators: numbers of significantly up-regulated miRNAs/total numbers of significantly differentiated expressed miRNAs. Black bars mean significantly down-regulated miRNAs. **b** Bar plots show percentages of overlapped genes between downregulated genes revealed by RNA-seq and predicted target genes of the upregulated miRNAs shown in (**a**) in *Dnmt*-null embryos. The top 50% high-confident targets of these miRNAs were chosen. Numerators/denominators: numbers of overlapped down-regulated genes/total numbers of downregulated genes shown in Supplementary Fig. 15a. **c** Enriched Gene Ontology terms of overlapped downregulated genes in E6.5-epi and E7.5-epi related to (**b**). *P* values were calculated by Metascape. **d** Representative genome browser snapshots of methylation profiles. The *IG*-DMR was divided into IG$^{CGI}$ (red-shaded box) and IG$^{TRE}$ (blue-shaded box). Resolution, 5kb. **e** Volcano plot shows the differently expressed *Dlk1-Dio3* miRNAs of DKO-AG-haESCs compared to WT-AG-haESCs[60]. **f** Volcano plot shows the differently expressed *Dlk1-Dio3* miRNAs of SC

embryos (E6.5 epiblast) from DKO-AG-haESCs compared to WT SC embryos. **g** Boxplot shows the expression of *Dlk1-Dio3* miRNAs between WT SC embryos and *Dnmt*-null SC embryos (E6.5 epiblast) produced by IMGZ. *P* values were determined by Student's two-sided paired *t*-test. **h** Representative images of E7.5 and E8.5 SC embryos carrying *Dnmt*-null. Control SC embryos were derived from DKO-AG-haESCs[60]. *Dnmt*-null SC embryos were derived by inactivation of *Dnmt1/3a/3b* in SC embryos through IMGZ. *Mir*−6KO;*Dnmt*-null SC embryos were generated by injection of DKO-AG-haESCs carrying six miRNA deletions into oocytes, followed by inactivation of *Dnmt1/3a/3b* by IMGZ. Scale bars, 100 μm in up panels, 200 μm in down panels. **i** Boxplot shows the expression of *Dlk1-Dio3* miRNAs between control SC embryos from DKO-AG-haESCs and SC embryos from *Mir*−6KO carrying *Dnmt*-null. *P*-values were determined by Student's two-sided paired *t*-test. **j** Representative images of RNA in situ hybridization of *T* probe in E7.5 SC embryos with different mutations. Scale bar, 200 μm. Numerators/denominators in **h** and **j**: numbers of SC embryos sustaining the phenotype shown in the image/total numbers of observed embryos. Source data are provided as a Source Data file.

mice also exhibits gastrulation failure due to the absence of miRNA caused by disruption of the miRNA biogenesis pathway[67]. Meanwhile, DGCR8-mediated global miRNA dosage controls embryonic germ layer specification of ESCs in vitro[68]. Our results provide in vivo evidence to support that DNA-methylation-mediated precise global miRNA dosage is also involved in fine-tuning gene expression, which might be involved in regulating gastrulation and germ layer specification.

## Methods

### Mice
All animal procedures were approved by the Institutional Animal Care and Use Committee of the Center for Excellence in Molecular Cell Science, Chinese Academy of Sciences. Mice were housed in individually ventilated cages (IVC) under a credited specific pathogen-free facility. The room has controlled temperature (20–25 °C), humidity (30-70 %) and light (12 h light-dark cycle). Zygotes are all collected from eight weeks female B6D2F1 (C57BL/6J♀× DBA2♂) mice, which were mated to ten-twenty weeks *Oct4-EGFP* males (C57BL/6J background). The ten weeks mT/mG males (full name B6.129(Cg)-Gt (ROSA)26Sor tm4(ACTB-tdTomato, -EGFP) Luo/J; JAX strain 007676) were used to evaluate the off-target effect of hA3A-eBE3-Y130F. The ten-fourteen weeks ICR females were used as pseudo-pregnant foster mothers. All embryos were collected and analyzed from recipients at specific developmental stages, including E6.5, E7.5, E8.5, E9.5, E10.5, and E11.5.

### In vitro transcription
The mRNA transcriptional templates of different base editors and *Cre* were amplified by PCR using KOD-Plus-Neo (TOYOBO) from plasmids pCMV-BE3 (Addgene#73021), BE4-Gam (Addgene#100806), pCMV-hA3A-BE3-Y130F (Addgene#113428), pCMV-hA3A-eBE3-Y130F (Addgene#113423), xCas9(3.7)-BE3 (Addgene#108380), xCas9(3.7)-BE4 (Addgene#108381) and pZ4f-Cre, purified by the Universal DNA Purification Kit (TIANGEN, Cat#DP214) and then transcribed using the mMACHINE T7 ULTRA Transcription Kit (Invitrogen, Cat#AM1345) following the manufacturer's instruction. The transcriptional templates of sgRNAs were amplified from pX330-mCherry (Addgene#98750) and transcribed in vitro using the MEGAshort-script T7 kit (Invitrogen, Cat#AM1354) following the manufacturer's instruction. mRNAs and sgRNAs were subsequently purified with the MEGAclear Transcription Clean-Up Kit (Invitrogen, Cat# AM1908), resuspended in hot (95 °C) RNase-free water and then stored at −80 °C.

### Zygote microinjection
Eight-week-old B6D2F1 females were superovulated with 6 international units of pregnant mare's serum gonadotropin (PMSG) for 48 h and then injected with 6 international units of human chorionic

gonadotropin (hCG), subsequently mated to homozygous *Oct4-EGFP* males for 12 h. Zygotes were harvested from oviducts of B6D2F1 females with a plug after 24 h post-hCG injection using hyaluronidase (Sigma, Cat# H3884). For zygote microinjection, the mixture of base editor mRNA (20 to 200 ng/μl) and sgRNA (100 ng/μl) was diluted in RNAase-free water, centrifuging at 4 °C, 13,400 g for 10 min, and then injected into the cytoplasm of zygotes in a droplet of HCZB medium containing 5 μg/ml cytochalasin B (CB, Sigma, Cat#C6762) using a micromanipulator (Olympus) and a FemtoJet microinjector (Eppendorf). The injected zygotes were cultured for 24 h to two-cell embryos and 96 h to blastocysts in AA-KSOM (Millipore, Cat#MR-106-D) medium at 37 °C under 5% CO$_2$ in the air. The two-cell embryos were transferred into the oviduct of pseudopregnant ICR females at 0.5 days postcopulation. The blastocysts were used for genotyping and morphological analysis.

### Blastocysts genotyping and Blunting cloning
Mouse blastocysts were directly lysed by 5 μl buffer from the Mouse Direct PCR Kit (Bimake, Cat# B40015), incubated at 55 °C for 50 min and 95 °C for 5 min. The targeted region was amplified by PCR using Phanta® Max Super-Fidelity DNA Polymerase (Vazyme, Cat#P505) from blastocyst lysis (about 2 μl), purified by gel electrophoresis using Universal DNA Purification Kit and then Sanger sequencing and Blunt cloning. The purified amplicons were cloned into pEASY-Blunt vector (TransGen Biotech, Cat#CB101), transformed into Trelief ™ 5α Chemically Competent Cell (Tsingke Biological Technology), and picked up more than 13 *E. coli* clones for sequencing.

### Intracytoplasmic sperm injection (ICSI) and two-cell embryo injection
For the ICSI experiment, MII oocytes were collected from oviducts of superovulated B6D2F1 females (eight-week-old) using hyaluronidase. Sperms obtained from homozygous mT/mG males (full name *B6.129(Cg)-Gt(ROSA)26Sor* $^{tm4(ACTB-tdTomato,-EGFP)Luo}$/J; JAX strain 007676) using H-CZB medium and sperm tails were broken by ultrasound using Ultrasonic Cleaner (KUDOS). The sperm head was injected into the cytoplasm of MII oocytes in a CB droplet using a Piezo-drill micromanipulator. The reconstructed zygotes were cultured in a CZB medium for 30 min and then activated for 6 h in Ca2$^+$-free CZB with SrCl$_2$. Following activation, all of the reconstructed embryos were cultured for 26 h in AA-KSOM medium to late-stage two-cell embryos. For two-cell embryo injection, the mixture of *Cre* mRNA (0.5 ng/μl), hA3A-eBE-Y130F mRNA (100 ng/μl), and sgRNA (100 ng/μl) was injected into the cytoplasm of one blastomere of the two-cell embryo derived from the ICSI experiment. The injected two-cell embryos were cultured in AA-KSOM medium at 37 °C under 5% CO$_2$ in the air for 1 h and then transferred into oviducts of pseudo-pregnant ICR females at 0.5 days post copulation.

## Flow cytometry

For GOIT, to isolate cells for whole-genome sequencing, E11.5 embryos were dissociated by 1 ml 0.25% Trypsin-EDTA (Gibco, Cat#25200056) in 37 °C water bath for 20–30 min, constantly blowing with a 1ml pipette tips. The digestion was terminated by adding 1 ml DMEM medium with 10% FBS (Excell Bio). The cell suspension was centrifuged at 300 g for 6 min, rinsed one time with DPBS, transferred through a 40-μm cell strainer, and tdtomato$^+$/EGFP$^+$ cells were enriched by sorting on BD FACS Aria II. More than one million cells per sample were collected for genome extraction.

## AG-haESCs culture, gene modification, and ICAHCI

A detailed protocol of AG-haESCs culture and intracytoplasmic AG-haESC injection (ICAHCI) has been described in previous works[59,60]. Briefly, mouse AG-haESCs (including WT- and DKO-AG-haESCs) were cultured in DMEM with 15% FBS (Excell Bio), penicillin–streptomycin, non-essential amino acids, NUC, l-glutamine, β-mercaptoethanol, 1000 U ml$^{-1}$ Lif, 1μM PD03259010 (Selleck) and 3μM CHIR99021 (Selleck). sgRNAs for different miRNA deletions were synthesized and ligated in pX330-mCherry plasmid[69]. Constructed plasmids were then transfected into O48, which were enriched haploid cells (Hoechst 33342 staining) and mCherry positive cells through FACS after 24 h. Gene-modified cell lines were obtained from single-cell expansion and genotyping, which were then subjected to the ICAHCI experiment.

## Western blot

Embryos were lysed in 1x RIPA buffer (Cell Signaling Technology, Cat#9806S) containing protease inhibitor cocktail (Sigma, Cat# P8340) on ice for 1 h. Before gel electrophoresis, protein lysate was boiled with 5x loading buffer at 100 °C for 10 min. The appropriate volume of protein lysate was loaded onto 8 % SDS-PAGE gels and separated by gel electrophoresis followed by transfer onto the PVDF membrane. Membranes were incubated with anti-DNMT1 (1:2000 dilution, Cell Signaling Technology, Cat#5032S), anti-DNMT3A (1:2000 dilution, Proteintech, Cat#20954-1-AP), anti-DNMT3B (1:2000 dilution, Abcam, Cat#ab2851) and anti-GAPDH (1:5000 dilution, Proteintech, Cat# 60004-1-Ig) at 4 °C overnight, followed by HRP-conjugated antirabbit (1:5000 dilution, BBI, Cat#D110058-0001) and anti-mouse antibody (1:10,000 dilution, Proteintech, Cat#SA00001-1) incubation for 1 h at room temperature.

## Whole-genome sequencing (WGS) and data analysis

Genomic DNA was extracted from cells (tdtomato$^+$ and EGFP$^+$, individually) by using TIANamp Micro DNA Kit (TIANGEN, Cat#DP316) according to the manufacturer's instructions. The libraries were constructed by NEBNext® Ultra™ II DNA Library Prep Kit for Illumina (NEB, Cat#E7645) and sequenced at an average depth of 40 X using the Illumina NovaSeq 6000 System at the Novogene Bioinformatics Institute, Beijing, China. We mapped the reads to the reference database (mm10) using BWA (v0.7.12). Then the BAM files were sorted and marked duplicates by Picard tools (v2.3.0). Three different algorithms, Mutect2 (v3.5), Lofreq (v2.1.2) and Strelka (v2.7.1) were separately used to call single nucleotide variation to ensure the high confidence of the genome-wide de novo variants found. At the same time, Mutect2 (v3.5), Scalpel (v0.5.3) and Strelka (v2.7.1) were run individually to detect the whole genome de novo indels. The overlap of three algorithms of SNVs or indels was considered as the true variants. The variants were identified in the mapped BAM file of samples with gene editing, using the samples without gene editing as control.

The WGS analysis revealed a low level of targeted editing in the control of Tyr and Crygc groups in the range of 0–4.7%. Thus, in the following analysis, we only keep the reliable variants with allele frequencies of more than 10%. We also removed variants that existed in the dbSNP (v138) and MGP (v3) databases. To validate the on-target efficiency, we also compared the aligned BAM files with the on-target

sites by pairwise Alignment function in the R package, Biostrings. We predicted the potential off-targets of targeted sites using Cas-OFFinder (http://www.rgenome.net/cas-offinder/) with all possible mismatches. The SNVs and indels were annotated with annovar (version 2016-02-01) using the RefSeq database. The off-target SNVs of all samples with gene editing were classified into transcribed and un-transcribed regions based on the annotation of RefSeq. Meanwhile, the same number of loci in the genome was simulated and calculated the proportion of the transcribed region. P-value was calculated with Fisher's exact test between the proportion of expected transcribed region in 10000 times simulation and the observed proportions of the transcribed region of the off-target SNVs.

## Base-editor tool

To facilitate the design of sgRNAs for the whole gene used for base editors, we deployed the Base-Editor tool, including server-side and client-side (https://github.com/DiabloRex/BASE-Editor). Base-Editor tool designs all possible BE sgRNAs for whole exons of a given gene, not specific DNA sequences and can efficiently design a sgRNA library and predict the resulting amino acids for base-editing screening, instead of the simple and time-consuming method in our previous work[16]. These tools integrated off-target searching and scoring, efficiency evaluation, and editing target indication. For core algorithms, to search the sgRNAs, we used the genome fast file and genome annotation file to locate the sequence for PAM selection. Then, those sgRNAs are subjected to local Cas-OFFinder (http://www.rgenome.net/cas-offinder/) for off-target analysis. Efficiencies are evaluated by using principles of machine learning models[70]. Currently, the server-side tool only supports mouse and human genomes, and requires only your gene name or NCBI Ref Seq ID. Two predefined editing modes are presented for the usual design (ABE and CBE editors). The client-side tool includes all the functions of the server-side tool, it also comes with multiple modifiable editing parameters and user-defined genomes for sgRNA design, and will be much faster when running on a client machine. The server-side tool is coded in C# ASP.NET language, and the client-side tool is implemented in C#.

## Targeted deep sequencing

Genomic DNA extracted from embryos (E6.5 to E10.5) using TIANamp Micro DNA Kit. Target sites were amplified from extracted genomic DNA using Phanta® Max Super-Fidelity DNA Polymerase (Vazyme). PCR products were ligated with different barcodes and then pooled together. The libraries were obtained from adapter-modified PCR and sequenced on Illumina HiSeq X Ten (2×150 PE) at Annoroad Gene Technology Corporation, Beijing, China. Supplementary Data 9 contains a list of primers used in this study.

## Bisulphite sequencing

Genomic DNA (about 100 ng) from the E9.5 embryo was treated with the EZ DNA Methylation-Direct Kit (ZYMO research, Cat# D5020) following the manufacturer's instructions. The recovery DNA products were subjected to amplify nested PCR. The bisulfite primers are listed in Supplementary Data 9. The amplified products were purified using the Universal DNA Purification Kit (TIANGEN) and cloned into a pMD19-T vector (Takara). For each sample, more than fifteen E. coli clones were picked up for stranded Sanger sequencing. The results were analyzed by the DNA methylation analysis tool (http://services.ibc.uni-stuttgart.de/BDPC/BISMA/).

## Whole-mount in situ hybridization

Embryos at E7.5 were dissected from the uteri and isolated from decidua in RNase-free PBS. Then the embryos were fixed in 4% paraformaldehyde overnight at 4 °C and dehydrated with different concentration gradients of methanol (25%, 50%, and 75%, diluted with PBST: PBS containing 0.1% Tween-20) and stored in 100% methanol at

−20 °C. Rehydrated embryos were prehybridized with a pre-hybridization solution at 70 °C for 3 h and then hybridized with digoxigenin-labeled riboprobes, which were generated by in vitro transcription from a linearized temple using a DIG RNA Labelling Kit (Roche, Cat#11175025910), at 70 °C overnight. Then embryos were blocked with a blocking solution containing Anti-DIG antibody conjugated to alkaline phosphatase (Roche, Cat#11093274910) overnight at 4 °C. The embryos were stained in 20 µl/ml NBT/BCIP (Roche, Cat#11681451001) at room temperature and then soaked in 80% gly-cerine until they sank to the bottom. Embryos were photographed on an OLYMPUS SZX16 microscope.

### Generation of *Dnmt* triple knockout embryos through mating

The strains of *Stra8-Cre* and *Zp3-Cre* mice used in this study were FVB/N-Tg (*Stra8-Cre*)1Reb32[71] and C57BL/6-TgN (*Zp3-Cre*) 93Knw33[72], respectively. *Dnmt1*^2lox, *Dnmt3a*^2lox and *Dnmt3b*^2lox mouse strains were described previously[73,74]. Owing to *Dnmt1* knockout will result in abnormal spermatogenesis, to inactivate all the three *Dnmt* genes in germ cells from the postnatal stage, we crossed *Dnmt1*^f/+/*Dnmt3a*^f/-*Dnmt3b*^f/-;*Stra8-Cre* males with *Dnmt1*^f/-/*Dnmt3a*^f/-*Dnmt3b*^f/-;*Zp3-Cre* females. *Stra8-Cre* mice express *Cre* in spermatogonia and *Zp3-Cre* mice express *Cre* exclusively in growing oocytes. In that way, 50% of offspring are *Dnmt*-null from the zygotic stage.

### LC-MS quantification of 5mC and 5hmC nucleosides

Purified genomic DNA was hydrolyzed enzymatically as described previously[75]. In brief, DNA was digested by nuclease P1 (Sigma, Cat#N8630) in the presence of 0.2 mM $ZnSO_4$ and 20 mM NaAc (pH 5.3) at 55 °C for at least 1 h and then was dephosphorylated with calf intestinal alkaline phosphatase (CIAP, Takara, Cat#2250A) at 37 °C for an additional 3 h. The samples were cleaned up by spin dialysis (Nanosep with 10K omega, PALL). The nucleosides were resolved by HPLC (Agilent 1260) as previously described[76]. In brief, a Hypersil Gold aQ column (Thermo Fisher Scientific, 100 mm; particle size, 1.9 µm) was used with a gradient elution starting with 0.1% (v/v) formic acid in water (A) and adding acetonitrile (LC-MS grade) with 0.1% (v/v) formic acid (B) at 25 °C. 0-5 min: 100% A, 5–12.5 min: 100% A to 88% A; 12.5–14 min: to 0 % A; 14–16 min: 100% A, 16–19 min: 100% A. The HPLC eluent was analyzed on an Agilent 6495 triple quadrupole mass spectrometer using multiple-reaction monitoring in positive ion mode. Transitions for each nucleoside: deoxycytidine (dC): 228.1→112.1 m/z, dG (Sigma, Cat#D7145), 5mdC: 242.1→126.1 m/z, 5hmdC: 258.1→142.1. A detailed procedure and instrumentation for the LC-MS/MS analyses are described previously[76]. For quantifications, a series of concentrations of nucleoside standards, were run for every batch of experiments to obtain their corresponding standard curves. dG (Sigma, Cat#D7145), dC (Sigma, Cat#D3897), 5mdC (Cayman, Cat#16166), 5hmdC (Berry & Associates, Cat#PY-7588). The ratios of 5mdC/dG and 5hmdC/C, were subsequently calculated. Data were analyzed using Microsoft Excel and GraphPad Prism 7 software.

### Genome-wide methylation sequencing

The DNA methylation library experimental procedure was mainly modified from the previously published method[77,78]. After extraction of mouse embryo genomic DNA, 10 ng of DNA was taken as the starting material for DNA methylation library preparation. 0.1 ng lambda DNA for each sample was spiked into the sample to quantify the C-to-T conversion efficiency. The bisulfite treatment was performed using EZ-96 DNA Methylation-Direct MagPrep (Zymo, Cat#D5044) based on user instruction, and the sample was eluted in 19.5 µL $H_2O$. To synthesize the first stand after bisulfite conversion, 2.5 µL 10× Blue buffer (TIANGEN), 2 µL 10 mM dNTP (TAKARA) and 1 µl 100 mM bio-P5-N6-Oligo1 (Z/5Biosg/CTACACGACGCTCTTCCGATCTNNNNNN) were added to the sample and the sample was incubated at 65 °C for 3 min

Then 1µL Klenow exo⁻ (TIANGEN) was added and the sample was incubated at 4 °C for 5 min, +1 °C/15 sec to 37 °C, 37 °C for 30 min The random priming was repeated 4 times and the free-primer was removed by treating with Exo I (NEB). The biotin-labelled fragments were enriched using Dynabeads M280 Streptavidin beads (Invitrogen) with rotation at room temperature for 45 min. The second strand was synthesized using P7-N9-oligo2 (AGACGTGTGCTCTTCC-GATCTNNNNNNN) and Klenow exo⁻ (TIANGEN). The final PCR was performed using NEB pre-indexed primer, universal primer, and KAPA HiFi HotStart ReadyMix (KAPA) for 5 cycles. 300-800 bp PCR production was isolated using 2 % agarose gel electrophoresis. Sequencing was performed on the Illumina Nova Seq 6000 at the Novogene Bioinformatics Institute, Beijing, China.

### Data processing of WGBS

Trim_galore (v0.6.4) (https://github.com/FelixKrueger/TrimGalore) was used to remove low-quality reads and adaptor sequences. Reads pass quality control were mapped to the mm10 mouse genome using BitMapperBS (v1.0.2.2) in pair-end mode with parameters "−sensitive–pbat–unmapped_out" at first. Unmapped reads were output by samtools (v1.11) and mapped again by using BitMapperBS's single-end mode. Aligned bam files were merged and sorted by samtools and the methylation information in MethylKit and bedGraph format were extracted by MethylDackel (v0.4.0) (https://github.com/dpryan79/MethylDackel) with default parameters. Bed format methylation files were generated by local scripts. The CpG sites coverage threshold used for DNA methylation estimation was 3X.

To calculate the global methylation level, the genome was divided into 1 kb tiles and the mean methylation levels of tiles (CpG sites number ≥ 3) were regarded as the global methylation level. To evaluate the reproducibility of our WGBS data, we used 1kb-tiled methylation levels to calculate Euclidean distance among all samples and performed hierarchical clustering, and the similarity between samples was defined as min-max normalized Euclidean distance reduced by 1. Genomic element annotations were downloaded from the UCSC genome browser. Promoters were defined as the 2 kb upstream to 2 kb downstream regions of the transcription start site (TSS), and the methylation levels of RefSeq gene bodies were calculated by the 'computeMatrix' function of deeptools (v3.4.3) in scale-regions mode. The average methylation levels of genomic elements were calculated using bedtools (v2.29.0), based on the methylation levels of those regions which covered at least 3 CpG sites. To compare our WGBS data with in vivo counterpart, data were downloaded from the GEO database with accession number (GSE76505[38]) and transformed to mm10 by UCSC binary tools 'liftOver'. Hierarchical clustering of global methylation levels of ours and Zhang et al. was performed by the 'hclust' function from R, and visualized using iTOL (https://itol.embl.de).

Differentially methylated regions were found based on genomic elements, such as promoter, enhancer, and transposable elements. The methylation levels of each genomic element in each sample were calculated as described above, then the hypermethylated elements were defined as elements with methylation levels difference greater than 30% and *P*-value less than 0.05 in multiple t-tests.

### RNA-seq and analysis

The embryos (E6.5 and E7.5) of control, *Dnmt1*-KO, *Dnmt3a/3b*-DKO, and *Dnmt*-null were carefully collected from mouse decidua. The epiblast (Epi) and extraembryonic ectoderm (Exe) of embryos were quickly separated by needle and obtained the cDNA library according to the reported protocol[61]. Sequencing was performed on the Illumina Nova Seq 6000 at the Novogene Bioinformatics Institute, Beijing, China. Trim galore (v0.6.4) was used to quality control for raw fastq data. After that, clean reads were mapped to the mm10 genome by STAR (2.7.2b) with parameters "-quant Mode

# Article

Transcriptome SAM Gene Counts" to output alignments translated into transcript coordinates in the bam files and read counts of each gene, which were then normalized to TPM (Transcript Per Millions) values. To ensure data quality, only replicates with Pearson's correlation greater than 0.9 were retained for further analysis. Gene-level count matrices were imported and performed differential expression analysis by R package DESeq2 (v1.26.0). We used the R function 'prcomp' to perform PCA (Principle Components Analysis). Gene ontology enrichment analysis was performed by online tools at Metascape (http://metascape.org).

## Retrotransposons expression analysis

Retrotransposons expression analysis was described in detail in a previous work[31]. In detail, unique and multiple mapped reads were assigned to class and family levels UCSC RepeatMasker annotation using featureCounts (v2.0.0) package with parameters '-M−fraction', which weight multimapping reads by total alignments. Differentially expressed retrotransposons were identified by DESeq2 (v1.26.0) with fold change > 2 and adjusted $P$ value < 0.05. To quantify the influence of retrotransposons on the expression of nearby genes, genes located within 20 kb upstream and downstream of TEs were considered.

## CAS-seq and analysis

The preparation of the CAS-seq cDNA library mainly followed the published protocol[45]. Total RNA was extracted from embryos (E6.5 and E7.5) and AG-haESCs by using miRNeasy Micro Kit (QIAGEN, Cat#217084) according to the manufacturer's instructions. The extracted RNA (100 pg) underwent the following steps to construct small RNA libraries: adapter ligation, first-strand cDNA synthesis, Cas9/sgRNA treatment, on-bead in vitro transcription, second RT, and PCR amplification, and gel purification. An Illumina NovaSeq 6000 System was used to sequence the small RNA libraries at the MingmaTechnologies Co., Ltd., China. The adapter sequences were removed from the 3' end of the sequenced reads using the fastp (version 0.20.1) with parameters "-l 15−length_limit 40". The remaining reads were mapped to the mouse genome (version GRCm38) using bowtie (version 1.2.1.1) first. The mapped reads were mapped again to the mature miRNA sequences annotated in miRbase (version 21), allowing −2 or +2 nt to be templated by the corresponding genomic sequence at the 3' end to determine the expression level of each miRNA. The differential expression analysis of each miRNA was performed by R package DESeq2 (version 1.30.1)[79]. Deseq2 first normalized miRNA count by calculating size factors, then estimated and shrunk miRNA dispersion, finally fitted Negative Binomial GLM and calculated $P$ value by Wald statistics. Since there are about 1300 expressed miRNAs (1972 annotated miRNAs in the mouse genome) in our sequencing data, to obtain more differentially expressed miRNAs, we cut off the $\log_2$ (FC)>0.5 and $P$ value <0.05 were defined as significantly different expressed miRNAs as used in previous[80].

## Statistical analysis

All experiments were independently repeated at least three times, giving similar results, and the findings of all key experiments were reliably reproduced. For the mouse experiments, we exceeded this sample size to confirm our results. The development and editing efficiency of this work was summarized in Supplementary Data 6. For all figures except for Supplementary Fig. 2f, the results shown are the average mean ± standard error of the mean (s.e.m). Statistical testing was performed using GraphPad Prism 7.

## Reporting summary

Further information on research design is available in the Nature Portfolio Reporting Summary linked to this article.

## Data availability

The whole genome sequencing data of E11.5 embryos data generated in this study have been deposited in the NCBI Sequence Read Archive database under accession code PRJNA574624. The deep sequencing data are deposited in the Gene Expression Omnibus (GEO) under the accession code GSE188658. The genome-wide methylation sequencing and RNA sequencing data from this study have been deposited in the GSE162903. The small RNA sequencing data have been deposited in the GSE205563. The original data presented in graphs generated in this study are provided in the Source Data file. Some source data with this paper and have been deposited in the figshare database and are available at https://doi.org/10.6084/m9.figshare.22580038. Source data are provided with this paper.

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

## Acknowledgements

We thank Prof. J. Chen (Shanghai Tech University) for providing pCMV-hA3A-BE3-Y130F and pCMV-hA3A-eBE3-Y130F plasmids and the help of the High-Performance Computing Platform of the Center for Life Sciences, Peking University. We thank Jane Li from UCB for critical reading of the manuscript. This study was supported by the Genome Tagging Project and grants from the National Key Research and Development Program of China (2019YFA0109900 and 2020YFA0509000 to J.Li.), the National Natural Science Foundation of China (31821004 to J.Li., 32030029 to J.Li., 32293231 to J.Li., and 3210060158 to Q.L.), and partly supported by Shanghai Municipal Science and Technology Major Project (22YS1400900 to J.Li.) and the Major Special Projects of Basic Research of Shanghai Science and Technology Commission (18JC1411103 to L.W.), and China Postdoctoral Science Foundation (2020M681407 to Q.L.). Q.L. gratefully acknowledges the support of Sanofi Scholarship Program.

## Author contributions

Q.L. and J.Li. conceived the project. Q.L., X.Y., Y.Che., and Y.W. carried out mouse embryo and cell studies, phenotype analysis, genotyping, and sample collection. C.W. and L.X. constructed *Dnmt* mutant embryos through mating, analyzed phenotypes, and performed LC-MS/MS. J.Lu., Y.G. and Q.L. constructed the WGBS library and analyzed the data of WGBS and RNA-seq. M.Y. and Q.L. developed the Base-Editor service and analyzed the deep-sequencing data. L.F., Y.S., R.Z. and Y.L. analyzed the off-target effect of the WGS data. Y.Cha., B.X., Y.Z. and L.W. constructed the small RNA library and analyzed these data. G.C., Q.L. and N.J. constructed the RNA-seq library. Q.L. and J.Li. designed the experiments and generated figures. Q.L. and J.Li. wrote the paper with the help of M.Y., L.W., F.T. and G.X. All authors read and approved the final manuscript.

## Competing interests

The authors declare no competing interests.

## Additional information

[1]State Key Laboratory of Cell Biology, Shanghai Key Laboratory of Molecular Andrology, Shanghai Institute of Biochemistry and Cell Biology, Center for Excellence in Molecular Cell Science, Chinese Academy of Sciences, University of Chinese Academy of Sciences, Shanghai, China. [2]School of Life Sciences, Biomedical Pioneering Innovation Center, Beijing Advanced Innovation Center for Genomics, Peking University, Beijing, China. [3]State Key Laboratory of Molecular Biology, Shanghai Key Laboratory of Molecular Andrology, Shanghai Institute of Biochemistry and Cell Biology, Center for Excellence in Molecular Cell Science, Chinese Academy of Sciences, Shanghai, China. [4]School of Life Science, Hangzhou Institute for Advanced Study, University of Chinese

Academy of Sciences, Hangzhou, China. [5]Institute of Neuroscience, CAS Center for Excellence in Brain Science and Intelligence Technology, Chinese Academy of Sciences, Shanghai, China. [6]CAS Key Laboratory of Systems Biology, Center for Excellence in Molecular Cell Science, Institute of Biochemistry and Cell Biology, Chinese Academy of Sciences, Shanghai, China. [7]Bio-Med Big Data Center, Key Laboratory of Computational Biology, CAS-MPG Partner Institute for Computational Biology, Shanghai Institute of Nutrition and Health, Chinese Academy of Sciences, Shanghai, China. [8]School of Life Science and Technology, Shanghai Tech University, Shanghai, China. [9]These authors contributed equally: Qing Li, Jiansen Lu, Xidi Yin, Yunjian Chang, Chao Wang, Meng Yan. ✉e-mail: glxu@sibcb.ac.cn; lgwu@sibcb.ac.cn; tangfuchou@pku.edu.cn; jsli@sibcb.ac.cn

