## [Peer Review File · Nature Communications]

Base editing-mediated one-step inactivation of the Dnmt gene family reveals critical roles of DNA methylation during mouse gastrulationEditorial Note: Parts of this Peer Review File have been redacted as indicated to remove third-party material where no permission to publish could be obtained.

REVIEWER COMMENTS

Reviewer #1 (Remarks to the Author):

In this manuscript, Li et.al. use a technique that they label Inactivation of Multiple Endogenous Genes in Zygotes (IMGZ) to simultaneously use base editing stop codons into multiple genes at the blastocyst stage of development. They demonstrate feasibility to delete all three Tet enzymes or all three Dnmt enzymes- Dnmt1, Dnmt3a and Dnmt3b. Dnmt-null embryos are unable to develop past gastrulation. They perform a through DNA methylation and gene expression profiling of the embryonic and extra embryonic tissue and find that the gastrulation block is due to misregulation of miRNAs and independent of retrotransposon methylation. Specifically, the lack of DNA methylation allows miRNA expression from the Dlk1-Dio3 which disrupts gastrulation. They dissect the regulation of the miRNA and find that while the IG-DMR suppresses Dlk1-Dio3 miRNA expression without affect the DNA methylation status, promoter methylation plays a partial role in inhibiting miRNA expression.

Major points

1)The IMGZ system is not novel and tests existing base editing methods.In fact the authors already cite the i-STOP method that has been used before. A rationale to use base editors to add a stop codon instead of just relying on Non-homologous end joining in blastocysts is also not articulated. Similarly the justification for a new method for choosing guides, why their "Base-editor" algorithm is better is not clear. A reference to Hwang, GH., Park, J., Lim, K. et al. Web-based design and analysis tools for CRISPR base editing. BMC Bioinformatics 19, 542 (2018). <https://doi.org/10.1186/s12859-018-2585-4>.

The IMGZ is the weakest portion of the paper and my suggestion would be to move it to the supplement so that the focus can be on the novel results with the triple embryo knockout of the Dnmt genes.

2) The simultaneous knockout of the three Dnmt proteins has not been performed during embryogenesis and provides the most novel findings. The authors find that the triple knockout does not survive through gastrulation due to dysregulation of miRNAs from the Dlk-Dio locus. Although they investigate the role of the IG-DMR vs promoter methylation in controlling expression of miRNAs- they do not differentiate that the IG-DMR itself is bipartite as found in a recent publication which is also not referenced.Aronson, B. E., Scourzic, L., Shah, V., Swanzey, E., Kloetgen, A., Polyzos, A., Sinha, A., Azziz, A., Caspi, I., Li, J., Pelham-Webb, B., Glenn, R. A., Vierbuchen, T., Wichterle, H., Tsigirigos, A., Dawlaty, M. M., Stadtfeld, M. & Apostolou, E. (2021). A bipartite element with allele-specific functions safeguards DNA methylation imprints at the Dlk1-Dio3 locus. *Developmental Cell*, 56(22), 3052-3065.e5. <https://doi.org/10.1016/j.devcel.2021.10.004>

In this paper DNA methylation of IG-DMR and its related control of Dlk1-Dio3 locus has been investigated with a viewpoint of its role in maternal or paternal imprinting. Given that the authors have the DNA methylation data in both the Dnmt TKO and the Tet TKO, they should dissect the methylation of the IG-DMR in each element separately and not treat it as a single entity.

3) Some data from the miRNA rescue experiment can be moved to the main manuscript from the supplement to strengthen their main conclusions of miRNA dysregulation in the Dnmt TKO.

4) The conclusion that DNA methylation controls gastrulation independent of TET mediated DNA demethylation is not surprising because if 5mC levels are already very low in the Dnmt-null embryos, then knockout of the TET enzymes after will not restore the 5mC to rescue the phenotype.

5) The suggested improvements are in the reanalysis of the IG-DMR to separate the bipartite element in the Scourzic et. al. paper.

Minor points

1)Missing reference for Zuo, E., Cai, Y.-J., Li, K., Wei, Y., Wang, B.-A., Sun, Y., Liu, Z., Liu, J., Hu, X., Wei, W., Huo, X., Shi, L., Tang, C., Liang, D., Wang, Y., Nie, Y.-H., Zhang, C.-C., Yao, X., Wang, X., ...

Yang, H. (2017). One-step generation of complete gene knockout mice and monkeys by CRISPR/Cas9-mediated gene editing with multiple sgRNAs. *Cell Research*, 27(7), 933–945. <https://doi.org/10.1038/cr.2017.81>.

- 2) Typos in the manuscript, some references to the figures in the text are incorrect
- 3) Abstract overstates that almost nothing is known about Dnmt role in embryogenesis. This statement can be softened to state that “while a lots is known, a triple knockout has not been examined”.

Reviewer #2 (Remarks to the Author):

Summary:

In this study, Li and co-authors present a novel system to CRISPR target loci in mouse zygotes. Using this approach, they generate and phenotype Dnmt-null embryos and Dnmt + Tet-null embryos. They report that Dnmt-null embryos die by E7.5 with failure to elongate the primitive streak. Dnmt-null embryos showed an upregulation of the miRNA cluster at the Dlk1-Dio3 locus, which the authors conclude is likely the molecular feature that underpins the early embryonic lethality.

The CRISPR-based base editing for in vivo genetic targeting presented in this study is impressively efficient and makes it possible to get generate combinations of knockouts that would be nearly impossible to acquire using conventional breeding methods. This represents an innovation in the field.

However, when it comes to the biological insights provided by this study through the combined targeting of DnmTs and/or Tets, I see some challenges in reconciling the findings presented here with the existing knowledge of the phenotypes and molecular features of single and double knockouts. Please find detailed comments below.

Major comments:

1. The validation of the CRISPR-based BE system is largely well-described (lines 91-111); however, one aspect that remains unclear is how frequent these random off-target mutations are in a given cell of a targeted embryo. The authors currently present this data in Extended data Fig. 2d as a total number of single nucleotide variants (SNVs) detected by whole genome sequencing (WGS) from E11.5 embryos. This is difficult to put into context because the total number of detected SNVs will depend on how deeply the WGS libraries were sequenced. Can the authors present these data as an estimated number of SNVs per thousand bases and extrapolate based on this, how many SNVs would be expected in each cell of a targeted embryo?
2. Related to the above point, the authors provide a detailed characterization of the sgRNA efficiencies and off-target mutations in the establishment of the method when using a single sgRNA (lines 91-126); however, it is likely that both of these parameters will change when multiplexing sgRNAs. Can the authors please provide additional information on efficiencies and off-target mutations when multiplexing CRISPR-based BE? This will be important for researchers who may want to apply this method in future.
3. Given the severe developmental impairment in the Dnmt-TKO embryos (Fig. 3), how did the authors validate that the dissections of the Epi and ExE in E6.5 and E7.5 embryos were as accurate as the controls? A concern may be that with such a small epiblast compartment in the Dnmt-TKO embryos, there may be variable amounts of cross-contamination of trophoblast cells and/or an over-representation of visceral endoderm in those datasets.
4. I appreciate the value of comparing data from the Dnmt-TKO to the Dnmt1 KO and/or Dnmt3a/3b DKO, as this can provide insights into what is uniquely dysregulated in the Dnmt-TKO. However, this

can often be confusing or misleading the text, such as statements like “Interestingly, we observed that germline-related genes were specifically down-regulated in both Dnmt1-KO and Dnmt3a/3b-DKO Epi of E7.5 when compared to Dnmt-null embryos” (lines 197-199). This sentence is misleading, as germline genes become de-repressed in both the Dnmt1 KO and Dnmt3a/3b DKO embryos (Dahlet et al. 2020 Nature Communications). Can the authors please revisit how some of the data is presented or discussed throughout the manuscript to improve clarity?

5. The generation of the Tet/Dnmt-6KO is intriguing, but the rationale, phenotypic comparisons and biological insights of these experiments could to be improved (lines 208-239):

- In the absence of 5mC substrate due to the lack of DNMTs in embryogenesis, can the authors explain further what insights the Tet/Dnmt-6KO provide about TET function(s)?
- The authors compare the phenotypes of Tet/Dnmt-6KO and Dnmt-KO embryos, concluding that “These [Tet/Dnmt-6KO] phenotypes are similar to Dnmt1/3a/3b-TKO embryos...” (lines 214-220). Can the authors please provide quantitative comparisons of the embryos and primitive streak formation to support this conclusion?
- To properly disentangle the roles and functions of DNMTs and TETs (lines 208-239), it seems essential to include Tet KO embryos. Tet TKO embryos die at gastrulation, so without including these embryos for comparison, I think it cannot be concluded that the Dnmt/Tet-6KO embryos are more like Dnmt-null rather than Tet TKOs. Previous reports demonstrated that loss of Dnmt3a or Dnmt3b in fact partially rescued embryogenesis the Tet TKO (Dai et al. 2016 Nature), so I think there are important biological insights that remain unexplored within this section.

6. The authors report that Dnmt-TKO shows profound developmental delay with impaired formation of the primitive streak and onset of gastrulation – indeed, this phenotype is much more severe than the mid-gestational lethality seen in the Dnmt1 KO or Dnmt3a/3b DKOs (Li et al. 1992 Cell; Okano et al. 1999 Cell), suggesting an importance of DNA methylation very early in post-implantation development. Yet, the authors observe remarkably few changes in the Dnmt TKO mRNA transcription and no evident correlation between loss of DNA methylation and gene expression changes. They report a de-repression of miRNAs, which are almost exclusively from the Dlk1-Dio3 imprinted cluster. The authors conclude that this underpins many of the gene expression and developmental changes seen in the Dnmt TKO. This proposed mechanism needs further exploration because there are several other knockout models where loss of imprinting at the Dlk1-Dio3 locus does not cause early lethality.

Dnmt1 KO embryos are unable to maintain inherited DNA methylation from the germline, and hence lose imprinted gDMRs, including the IG-DMR at the Dlk1-Dio3 locus (as confirmed in this study – lines 339-344). DNA methylation at the IG-DMR controls the imprinting of the Dlk1-Dio3 miRNA cluster (Seitz et al. 2004 Genome Res); hence, Dnmt1 KO embryos fail to repress the paternal allele, including the miRNA cluster (Extended data Fig 15b). Parthenogenetic embryos are generated through the activation of an oocyte; consequently, with two maternal alleles, these embryos would also express the Dlk1-Dio3 miRNA cluster from both alleles. However, both Dnmt1 KO and parthenogenetic embryos die at mid-gestation (Li et al. 1992 Cell; Surani and Barton. 1983 Science). The authors need to reconcile why the de-repression of the Dlk1-Dio3 miRNA cluster in the context of the Dnmt1 KO and parthenogenetic embryos does not impact gastrulation, while in the Dnmt TKO it does so profoundly. In particular, given the finding that Dlk1-Dio3 miRNA upregulation was remarkably similar between Dnmt1 KO and Dnmt-TKO embryos (lines 335-339, Extended data Fig 15b).

Comments:

1. The mechanism that underpins active demethylation during embryogenesis remains contentious, as TET3-mediated demethylation does not appear to be the primary mechanism (Amouroux et al. 2016 Nat Cell Biol), as described in lines 57-61. Please revise the phrasing of this sentence.
2. The statement “these strategies can’t exclude the potential influences of the family genes in gametogenesis, and thus can’t distinguish the roles of gametogenesis and/or embryogenesis” (lines

71-73) is over-stated. Conventionally, heterozygous constitutive knockout animals are used to derive knockout embryos, and germline conditional knockouts are used to generate gamete-specific knockouts; these strategies robustly differentiate between the effects of a gene in gametogenesis or embryogenesis. Please amend or delete this statement.

3. Can the authors please check the images in Figure 3e – it appears that the images for Dnmt1 KO and the Dnmt-TKO have been switched?

4. I disagree with the conclusions that are based on the data presented in Fig 4c (lines 226-229). The higher amount of 5mC observed in the Tet/Dnmt3a/3b-5KO than Tet/Dnmt1-4KO E8.5 embryos does not necessarily indicate that methylation established by de novo DNMTs are vulnerable to TETs because the absence of DNMT1 means that any established de novo methylation will not be maintained. Indeed, in Fig 4e, the levels of DNA methylation in the Tet/Dnmt3a/3b-5KO appear similar to those in the Dnmt3a/3b DKO.

5. The phrasing around the findings presented in Extended data Fig 8a, "... Dnmt1-KO samples were much closer to late development stages of control embryos" (lines 249-250) is somewhat misleading. The Dnmt1-KO embryos cannot maintain DNA methylation, so the apparent 'similarity' between Dnmt1-KO embryos and E6.5/E7.5 embryos is a transient one, which will be lost once the de novo DNMTs are no longer expressed. Please consider revising this statement.

6. Can the authors ensure that the same terminology for the knockouts is used throughout the manuscript? For example, Dnmt-null and Dnmt-TKO are used interchangeably.

7. The authors report that Dnmt3a/3b DKO embryos show higher levels of DNA methylation than blastocysts (lines 266-269, Extended data Fig 9), concluding that DNMT1 has de novo methyltransferase function, similar to recent reports (Haggerty et al. 2021 Nature). However, the findings reported in this study are not consistent with that reported by Haggerty et al. The previous study found evidence for de novo activity of DNMT1 at a subset of repetitive elements in the absence of DNMT3A and 3B; whereas, the findings reported here show a significant gain of DNA methylation in Dnmt3a/3b DKO embryos across genomic features. The discussion of these results should be amended.

8. Can the authors further explain why there are such dramatic differences in gene expression between the Dnmt-TKO E6.5 epiblast versus controls, but not in E7.5 epiblast (Extended data Fig 11e)? Given the dramatic phenotype and lack of effective gastrulation in the Dnmt-TKO embryos, the gene expression differences at E7.5 should be substantial.

9. ESCs can have high levels of genomic instability in 2i conditions. The culture of haploid ESCs in 2i conditions in combination with CRISPR targeting could result in vast genomic instability (lines 360-399). It should be confirmed that the genome of these cells is at least somewhat intact.

10. Can the authors further explain the use of androgenetic haploid (AH) ESCs to study the Dlk1-Dio3 imprinted domain? As the miRNA cluster at this locus is expressed from the maternal allele, why would deletion of the IG-DMR on the paternal allele result in down-regulation of the miRNA cluster in AH-ESCs (lines 368-372, Extended data Fig 17g)?

11. Can the authors add justification for using >1.414 fold change for the miRNA analysis? Can you please also clarify whether the p<0.05 cutoff for significance was corrected for multiple comparisons? (lines 1180-1182)

Reviewer #3 (Remarks to the Author):

This is a very exciting study investigating the role of DNA methylation during gastrulation using dCas9-AID base editors to simultaneously inactivate DNMT1,3a and 3b. Simultaneous inactivation of genes is very difficult with mouse breeding approaches, and in particular for DNA methylation, many confounding effects would make interpretation very difficult.

The authors tested different base editors (BE), and used the most efficient one to induce triple KO embryos for Tet1. The frequency for homozygous premature mutation was quite high, 56%.

Quite extraordinary is that they manage a 6x mutant (all three DNMTs and all three TETs - simultaneously) – the efficiency was 1 in 6 embryos having all 6 genes mutated!

The IMGZ technology offered a number of insights, some confirmation of previous knowledge, some novel. Among the novel findings, they show evidence that DNA methylation of retrotransposons and enhancers may not be critical for gastrulation. Especially the latter is quite interesting and is unexpected, but they find enhancers relevant for organ and tissue development.

The manuscript was exciting from the beginning but got better and better with every figure. They find that DNA methylation primarily suppresses miRNAs (46.7% of targets) and protein coding genes, but a lower fraction (28%). The majority of the 54 miRNA were in the Dlk1-Dio3 region, and they primarily targeted downregulation of genes that were found downregulated upon DNMT KO.

Can the authors speculate what would the effects be if they generated catalytic mutants instead of full KOs? In other words, can one separate catalytic from non-catalytic roles for DNMTs and TETs during development?

Comments:

- the authors developed a website to design sgRNAs and predict off-target effects. Is this website publicly available? What algorithm does it use?
- in Fig 2f, DNA methylation was measured by mass spec and the value 5mC/C is 15-20%. Normally this is measured over G and in somatic cells (fully methylated genome) this value is 4%, why do the authors measure it over C?
- What is the embryo attrition in these experiment? From untreated to injected with no DNA, from no DNA injection to injection with only the BE (and separately for sgRNA injection), and between injection on BE+sgRNA with and without catalytic activity? The same question for DNMT experiments
- Line 162 directs to Suppl Fig 4c – should it be Suppl Fig 4d? Also, the authors write that "...suggesting that DNMT enzymes may be not essential for the egg cylinder formation"- since DNA methylation is measured at E8.5 (complete loss) and DNMT genes are inactivated in zygotes, could there be that there is a leftover DNA methylation at implantation that gives the phenotypes at E7.5? Is there evidence in the literature that also indicated that DNA methylation is not essential for the onset of gastrulation?
- Line 269, yes and this study shows the same (Haggerty, Meissner PMID: 34140676)
- Line 281 should be "embryos only slightly influenced" instead of "embryos slightly influenced" since it's a comparison to the above Extended Fig 10c
- Line 325, in "down-regulated genes" you mean protein coding genes, correct?

The response to reviewers' comments:

Reviewer #1 (Remarks to the Author):

In this manuscript, Li et.al. use a technique that they label Inactivation of Multiple Endogenous Genes in Zygotes (IMGZ) to simultaneously use base editing stop codons into multiple genes at the blastocyst stage of development. They demonstrate feasibility to delete all three Tet enzymes or all three Dnmt enzymes- Dnmt1, Dnmt3a and Dnmt3b. Dnmt-null embryos are unable to develop past gastrulation. They perform a through DNA methylation and gene expression profiling of the embryonic and extra embryonic tissue and find that the gastrulation block is due to misregulation of miRNAs and independent of retrotransposon methylation. Specifically, the lack of DNA methylation allows miRNA expression from the Dlk1-Dio3 which disrupts gastrulation. They dissect the regulation of the miRNA and find that while the IG-DMR suppresses Dlk1-Dio3 miRNA expression without affect the DNA methylation status, promoter methylation plays a partial role in inhibiting miRNA expression.

Response: We greatly appreciate the reviewer's recognition of the significance of our work.

Major points

1)The IMGZ system is not novel and tests existing base editing methods. In fact the authors already cite the i-STOP method that has been used before. A rationale to use base editors to add a stop codon instead of just relying on Non-homologous end joining in blastocysts is also not articulated. Similarly the justification for a new method for choosing guides, why their "Base-editor" algorithm is better is not clear. A reference to Hwang, GH., Park, J., Lim, K. et al. Web-based design and analysis tools for CRISPR base editing. BMC Bioinformatics 19, 542 (2018). <https://doi.org/10.1186/s12859-018-2585-4>.

The IMGZ is the weakest portion of the paper and my suggestion would be to move it to the supplement so that the focus can be on the novel results with the triple embryo knockout of the Dnmt genes.

Response: Thanks for these criticisms. We agree that the i-STOP method is an open idea to inactivate endogenous genes by cytosine base editing (BE) in the field of genome editing, which has been successfully applied in cultured cells¹. Meanwhile, the BE3, the original version of the base editor, has also been used to disrupt multiple genes in mouse embryos combined with two sgRNAs for each gene; however, the mutations of the targeted genes are complicated in both alleles², limiting its applications in studying the function of multiple genes in embryonic development. Given that both base editor (BE) and sgRNA determine the editing efficiency of C-to-T conversion^{3,4}, it is critical to select most efficient BE and sgRNA for one-step inactivation of multiple genes in zygotes efficiently. Therefore, in the beginning this study, we compared several available BEs, including E3, Gam-BE4, hA3A-BE3-Y130F, hA3A-eBE3-Y130F, X-BE3, and X-BE4 and found that hA3A-eBE3-Y130F was led to the most efficient C to T conversion (~90%) in the resultant blastocysts among all tested BE systems, with a higher ratio of homozygous mutation and fewer side-products.

Meanwhile, to select the most efficient sgRNA, we developed "Base-editor" algorithm to design all possible BE sgRNAs for whole exons of a given gene. We thank the reviewer for raising the issue about the available web tool of BE-Designer developed by Hwang et al., which provides all possible base editor sgRNAs in a given input DNA sequence with useful information including

potential off-target sites⁵. However, it was not published when we developed the “Base-Editor”.

Moreover, our Base-Editor tool designs all possible BE sgRNAs for all exons of a given gene, not specific DNA sequences, can efficiently design a sgRNA library, and predict the resulting amino acids for base-editing screening. We have added the relevant description and discussion in the Methods section about the Base-Editor tool. In addition, the mutation results of NHEJ are complicated and unpredictable compared to base editing, including various insertions and deletions, some of which are 3n base pairs that cannot achieve frameshift mutations and gene knockout. The i-STOP method mediated by base editing is more efficient and controllable, and is an effective alternative to precisely generate gene knockout. We have added the relevant description and discussion in the Methods section.

We appreciate the reviewer’s suggestion for the rearrangement of our figures. We have now merged main Figures 1 and 2 into Figure 1 of the revised manuscript and moved some data to the supplementary materials. Besides, to strengthen the advantages of the IMGZ system, we added the phenotypes of *Tet*-TKO embryos obtained by the IMGZ to revised Figure 1 and moved the RNA-seq results of *Dnmt* mutants from the supplementary material to revised Figure 3.

2) *The simultaneous knockout of the three Dnmt proteins has not been performed during embryogenesis and provides the most novel findings. The authors find that the triple knockout does not survive through gastrulation due to dysregulation of miRNAs from the Dlk-Dio locus. Although they investigate the role of the IG-DMR vs promoter methylation in controlling expression of miRNAs- they do not differentiate that the IG-DMR itself is bipartite as found in a recent publication which is also not referenced. Aronson, B. E., Scourzic, L., Shah, V., Swanzey, E., Kloetgen, A., Polyzos, A., Sinha, A., Azziz, A., Caspi, I., Li, J., Pelham-Webb, B., Glenn, R. A., Vierbuchen, T., Wichterle, H., Tsirigos, A., Dawlaty, M. M., Stadfeld, M. & Apostolou, E. (2021). A bipartite element with allele-specific functions safeguards DNA methylation imprints at the Dlk1-Dio3 locus. Developmental Cell, 56(22), 3052-3065.e5. <https://doi.org/10.1016/j.devcel.2021.10.004>)*

In this paper DNA methylation of IG-DMR and its related control of Dlk1-Dio3 locus has been investigated with a viewpoint of its role in maternal or paternal imprinting. Given that the authors have the DNA methylation data in both the Dnmt TKO and the Tet TKO, they should dissect the methylation of the IG-DMR in each element separately and not treat it as a single entity.

Response: We appreciate the reviewer’s suggestion. Aronson et al. reported that the IG-DMR consists of two antagonistic elements: a paternally methylated CpG island (IG^{CGI}) that prevents recruitment of TET dioxygenases and a maternally unmethylated non-canonical enhancer (IG^{TRE}) that ensures expression of the *Gtl2* lncRNA by counteracting de novo DNA methyltransferase⁶. The aberrant hypermethylation of maternal IG^{CGI} upon maternal IG^{TRE} deletion, whereas the imprint remains stable after the paternal IG^{TRE} deletion, indicating that IG^{TRE} deletion just like IG-DMR deletion can maintain the imprint of *Dlk1-Dio3* locus in bi-maternal and semi-cloned embryos⁶⁻⁸. Thus, we reanalyzed the methylation changes of IG^{CGI} and IG^{TRE} in different mutant embryos and found that DNA methylation of IG^{CGI} and IG^{TRE} was sustained in *Dnmt3a/3b*-DKO embryos, but was lost in *Dnmt1*-KO and *Dnmt*-null embryos (Fig. 6d). However, the methylation status of IG^{CGI} and IG^{TRE} showed no difference in *Tet*-TKO embryo compared to controls (Fig. R1a). Together, these results indicate that the methylation status of both IG^{CGI} and IG^{TRE} is controlled by DNMT1, consistent with our observations in the first version of manuscript and a previous report⁹.

Our results showed that deletion of *IG-DMR* in androgenetic haploid embryonic stem cells (AG-haESCs) can suppress the miRNA expression of the *Dlk1-Dio3* locus in AG-haESCs (Extended Data Fig. 17g). As the reviewer suggested, we next attempted to dissect the influence of IG^{TRE} on the miRNA expression in the *Dlk1-Dio3* locus and *IG-DMR*. To this, we used CRISPR-Cas9 technology to delete IG^{TRE} or *IG-DMR* (Fig. R1b) in a previously reported AG-haESC line carrying *H19^{ΔDMR}* (termed *H19^{ΔDMR}*-AGH cell line, HG125)⁸. We analyzed the methylation status of the IG^{CGI} and found that deletion of IG^{TRE} definitely can remethylate the IG^{CGI} (Fig. R1c, d), which is similar to the reported result⁶. The small RNA-sequencing also suggested that deletion of IG^{TRE} can suppress the *Dlk1-Dio3* miRNAs compared to HG125 AG-haESCs, but the degree of suppression is lower than *IG-DMR* deletion (Fig. R1e, f). These results suggest that IG^{TRE} methylation alone can also suppress the expression of paternal *Dlk1-Dio3* miRNAs.

Fig. R1 | IG^{TRE} methylation alone can suppress *Dlk1-Dio3* miRNAs.

a, Representative genome browser snapshots of methylation profiles in *IG-DMR* of control and seven mutant Epi at E7.5. The *IG-DMR* was divided into IG^{CGI} (indicated by red-shaded box) and IG^{TRE} (indicated by blue-shaded box). Resolution, 2kb. **b**, Schematic illustration of constructing $IG^{\Delta TRE}$ and $IG^{\Delta DMR}$ AG-haESCs by CRISPR-Cas9 technology. **c**, Methylation analysis of CGI-1 and CGI-2 of the IG^{CGI} region in HG125 and $IG^{\Delta TRE}$ AG-haESCs. Open circles represent unmethylated CpG sites, whereas filled circles represent methylated CpG sites. **d**, DNA methylation levels of CGI-1 and CGI-2 regions in HG125 and $IG^{\Delta TRE}$ AG-haESCs. Data are mean \pm s.e.m for three biological replicates. ** $P < 0.01$; *** $P < 0.001$ by Student's unpaired two-sided *t*-test. **e**, Volcano plot shows the differentially expressed miRNAs of $IG^{\Delta TRE}$ vs HG125 AG-haESCs, $IG^{\Delta DMR}$ vs HG125 AG-haESCs, and $IG^{\Delta DMR}$ vs $IG^{\Delta TRE}$ AG-haESCs. **f**, Box plot shows the expression levels of *Dlk1-Dio3* miRNAs in HG125, $IG^{\Delta TRE}$, and $IG^{\Delta DMR}$ AG-haESCs. *** $P < 0.001$ by Student's unpaired two-sided *t*-test.

haESCs, and IG^{ΔDMR} vs IG^{ΔTRE} AG-haESCs in the *Dlk1-Dio3* miRNAs. Significance was calculated by Deseq2; Log2 (fold change) > 0.5; *P*-value < 0.05. **f**, Bar plot shows the expression of *Dlk1-Dio3* miRNAs in HG125, IG^{ΔTRE}, and IG^{ΔDMR} AG-haESCs. Significance was determined by Student's paired t-test.

3) *Some data from the miRNA rescue experiment can be moved to the main manuscript from the supplement to strengthen their main conclusions of miRNA dysregulation in the Dnmt TKO.*

Response: We appreciate the reviewer's suggestion, and we have moved the methylation data of IG-DMR and E8.5 phenotype from the supplement figure to the main figure (Fig. 6).

4) *The conclusion that DNA methylation controls gastrulation independent of TET mediated DNA demethylation is not surprising because if 5mC levels are already very low in the Dnmt-null embryos, then knockout of the TET enzymes after will not restore the 5mC to rescue the phenotype.*

Response: Thank you very much for your comment. We agree that the opinion that the function of DNA methylation is independent of TET mediated DNA demethylation is not a surprising concept per se in the DNA methylation/demethylation field. Rather, it represents a prevailing opinion that is generally accepted by most of the experts in the field. However, despite the fact that it is a prevailing hypothesis, to the best of our knowledge, it has not yet been directly tested experimentally. Scientific hypothesis is an educated guess, based on observation. However, it needs to be supported with repeated testing to become a theory. In this regard, nothing is better than a direct, heads-on test of the hypothesis. Sometimes, a hypothesis can be refuted through additional experimentation. For example, the hypothesis that odorant receptor gene choice is controlled by DNA rearrangements in olfactory sensory neuron existed for more than 10 years before two independent reports in Nature^{10,11} firmly conclude that this is actually not the case, through a direct testing of the central point of this hypothesis. Therefore, it is important to test this hypothesis experimentally. In this study, we established IMGZ system to generate *Tet/Dnmt1-4KO*, *Tet/Dnmt3a/3b-5KO*, and *Tet/Dnmt-6KO* embryos and found that these embryos exhibit similar phenotypes shown in *Dnmt1-KO*, *Dnmt3a/3b-DKO*, and *Dnmt-null* embryos, respectively. Our results provide the most direct evidence to date in support of the opinion that DNA methylation function is independent of TET-mediated DNA demethylation.

5) *The suggested improvements are in the reanalysis of the IG-DMR to separate the bipartite element in the Scourzic et. al. paper.*

Response: We appreciate the reviewer's suggestion. We have reanalyzed and performed additional experiments to dissect the function of bipartite elements in IG-DMR and modified the manuscript accordingly.

Minor points

1) *Missing reference for Zuo, E., Cai, Y.-J., Li, K., Wei, Y., Wang, B.-A., Sun, Y., Liu, Z., Liu, J., Hu, X., Wei, W., Huo, X., Shi, L., Tang, C., Liang, D., Wang, Y., Nie, Y.-H., Zhang, C.-C., Yao, X., Wang, X., ... Yang, H. (2017). One-step generation of complete gene knockout mice and monkeys by CRISPR/Cas9-mediated gene editing with multiple sgRNAs. Cell Research, 27(7), 933–945. <https://doi.org/10.1038/cr.2017.81>.*

Response: Thank you for this suggestion. We have added this citation in the introduction section of

the revised manuscript.

2) *Typos in the manuscript, some references to the figures in the text are incorrect*

Response: We are sorry about these mistakes. We have corrected them in the revised manuscript, such as Figure 3e and Extended Data Fig. 4c in the original version.

3) *Abstract overstates that almost nothing is known about Dnmt role in embryogenesis. This statement can be softened to state that “while a lots is known, a triple knockout has not been examined”.*

Response: We agree with the reviewer’s view. We have amended this description to “While a lot is known, the functional significance of DNA methylation in embryogenesis remains to be revealed by generating the *Dnmt*-null embryos.” in the revised manuscript.

Reviewer #2 (Remarks to the Author):

Summary:

In this study, Li and co-authors present a novel system to CRISPR target loci in mouse zygotes. Using this approach, they generate and phenotype Dnmt-null embryos and Dnmt + Tet-null embryos. They report that Dnmt-null embryos die by E7.5 with failure to elongate the primitive streak. Dnmt-null embryos showed an upregulation of the miRNA cluster at the Dlk1-Dio3 locus, which the authors conclude is likely the molecular feature that underpins the early embryonic lethality.

The CRISPR-based base editing for in vivo genetic targeting presented in this study is impressively efficient and makes it possible to get generate combinations of knockouts that would be nearly impossible to acquire using conventional breeding methods. This represents an innovation in the field.

Response: We greatly appreciate the reviewer’s comments.

However, when it comes to the biological insights provided by this study through the combined targeting of Dnmts and/or Tets, I see some challenges in reconciling the findings presented here with the existing knowledge of the phenotypes and molecular features of single and double knockouts. Please find detailed comments below.

Major comments:

1. *The validation of the CRISPR-based BE system is largely well-described (lines 91-111); however, one aspect that remains unclear is how frequent these random off-target mutations are in a given cell of a targeted embryo. The authors currently present this data in Extended data Fig. 2d as a total number of single nucleotide variants (SNVs) detected by whole genome sequencing (WGS) from E11.5 embryos. This is difficult to put into context because the total number of detected SNVs will depend on how deeply the WGS libraries were sequenced. Can the authors present these data as an estimated number of SNVs per thousand bases and extrapolate based on this, how many SNVs would be expected in each cell of a targeted embryo?*

Response: We greatly appreciate the reviewer’s questions. To address the off-target effects induced by hA3A-eBE3-Y130F, we employed the GOIT (Genome-wide off-target analysis by two-cell embryo injection) system developed by Zuo et al^{12,13}. Briefly, we injected Cre mRNA together with

hA3A-eBE3-Y130F mRNA (with or without sgRNA) into one blastomere of the two-cell embryo obtained from intracytoplasmic injection of mT/mG sperm (Extended Data Fig. 2a), followed by collection of derivative cells of edited (EGFP positive) and non-edited (tdTomato positive) blastomeres at embryonic day 11.5 (E11.5) for whole genome sequencing with an average depth of 40 X (the mouse genome size is about 2.8 Gb) (Extended Data Fig. 2b). The deamination by hA3A-eBE3-Y130F happened from the 2-cell to the 16-cell stage, when mRNA was injected into one blastomere of the two-cell embryo¹². Our results showed that hA3A-eBE3-Y130F induced substantial sgRNA-independent off-target SNVs, which could be produced by the random effects of the BE in one or more cells in the early embryo stage, consistent with the previous observations¹². In addition, to improve the presentation, as the reviewer suggested, we showed the number of SNVs per thousand bases in all chromosomes, which display an almost even distribution in genome (Extended Data Fig. 3a), suggesting that these off-target SNVs are randomly produced by hA3A-eBE3-Y130F.

2. Related to the above point, the authors provide a detailed characterization of the sgRNA efficiencies and off-target mutations in the establishment of the method when using a single sgRNA (lines 91-126); however, it is likely that both of these parameters will change when multiplexing sgRNAs. Can the authors please provide additional information on efficiencies and off-target mutations when multiplexing CRISPR-based BE? This will be important for researchers who may want to apply this method in future.

Response: We greatly appreciate the reviewer's suggestions. Regarding the editing efficiency of multiplexing sgRNAs, we have summarized the efficiency of homozygous mutations of embryos generated by IMGZ in Fig. 4f and Supplementary Table 5 of the revised manuscript. As you can see, the overall efficiency of homozygous mutations in multiple genes are decreased with increased number of sgRNAs. For the off-target effects induced by multiplexing sgRNA, we performed additional GOIT experiments by injection of hA3A-eBE3-Y130F mRNA with *Tyr*-sgRNA, *Crygc*-sgRNA, and *Dnmt3a*-sg3 into one blastomere of 2-cell embryos. WGS analysis showed that the number of SNVs and indels from multiple sgRNAs are similar to single sgRNA, indicating that the off-target effects in IMGZ embryos are sgRNA-independent and randomly occurred by the activity of deaminase (Fig. 1f, g and Extended Data Fig. 2d).

3. Given the severe developmental impairment in the Dnmt-TKO embryos (Fig. 3), how did the authors validate that the dissections of the Epi and ExE in E6.5 and E7.5 embryos were as accurate as the controls? A concern may be that with such a small epiblast compartment in the Dnmt-TKO embryos, there may be variable amounts of cross-contamination of trophoblast cells and/or an over-representation of visceral endoderm in those datasets.

Response: We appreciate the reviewer's concerns. We also noticed that the morphology impairment in the *Dnmt*-TKO embryos may affect sampling and the accuracy of sequencing data. In order to exclude the potential influences caused by sampling, we used embryos carrying *Oct4-EGFP* transgene in this study to better dissect the extraembryonic ectoderm (Exe, EGFP-negative) and epiblast (Epi, EGFP-positive) of E6.5 and E7.5 embryos (Fig. R2a, b). Meanwhile, the Exe is the part morphologically different from the ectoplacental cone and Epi, and there is a clear boundary between the Exe and Epi (Fig. R2a, b). Moreover, all Exe and Epi samples may not include the outer trophoblast cells, which were surgically stripped (Fig. R2a, b). Therefore, these samples are unlikely

to have cross-contamination with trophoblast cells.

The murine visceral endoderm (VE) is an extraembryonic cell layer originating from inner cell mass (ICM) that appears prior to gastrulation¹⁴. Given that the layer of VE is small and surround the Epi and Exe, we kept VE cells as a part of Exe and Epi (Fig. R2a, b). To exclude the potential over-representation of visceral endoderm in our datasets, we analyzed the epiblast marker genes and found that the expressional levels are higher in *Dnmt*-null Epi (Extended Data Fig. 6). Meanwhile, the principal component analysis (PCA) of transcriptome data showed that all Exe samples were clustered together, but separated from Epi samples which were also exclusively clustered together, indicating that the transcriptome differences between all samples are mainly caused by their different lineage origins (Fig. 3b). Furthermore, we analyzed the top 500 up-regulated and down-regulated genes identified between the control Exe and Epi among different samples. The results showed expressional patterns of Exe and Epi in mutant embryos (*Dnmt1*-KO, *Dnmt3a/3b*-DKO, and *Dnmt*-null) were similar to control samples (Fig. R2c, d). Together, these results indicate that a small part VE cells in Exe and Epi samples may not influence our datasets.

Fig. R2| Samples collection of E6.5 and E7.5 embryos.

a, b, Representative images of IMGZ-derived control and *Dnmt*-null embryos at E6.5 (**a**) and E7.5

(b). Oct4-EGFP signaling indicates the epiblast region. The whole embryo consists of ectoplacental cone, extraembryonic ectoderm (Exe), and epiblast (Epi). Scale bar, 200 μ m. **c, d**, The heatmaps show the expression of top 500 up- and down-regulated DEGs (identified between the control Epi and Exe at E6.5 and E7.5) in *Dnmt1*-KO, *Dnmt3a/3b*-DKO, and *Dnmt*-null Epi and Exe at E6.5 (c) and E7.5 (d). These results show that the transcriptome differences of Epi and Exe in different mutants are mainly caused by their lineage origins.

4. I appreciate the value of comparing data from the *Dnmt*-TKO to the *Dnmt1* KO and/or *Dnmt3a/3b* DKO, as this can provide insights into what is uniquely dysregulated in the *Dnmt*-TKO. However, this can often be confusing or misleading the text, such as statements like “Interestingly, we observed that germline-related genes were specifically down-regulated in both *Dnmt1*-KO and *Dnmt3a/3b*-DKO Epi of E7.5 when compared to *Dnmt*-null embryos” (lines 197-199). This sentence is misleading, as germline genes become de-repressed in both the *Dnmt1* KO and *Dnmt3a/3b* DKO embryos (Dahlet et al. 2020 Nature Communications). Can the authors please revisit how some of the data is presented or discussed throughout the manuscript to improve clarity?

Response: We apologize for these misunderstandings. We have represented these sentences in the revised manuscript. For example, we changed the sentence “Interestingly, we observed that germline-related genes were specifically down-regulated in both *Dnmt1*-KO and *Dnmt3a/3b*-DKO Epi of E7.5 when compared to *Dnmt*-null embryos” to “Besides, we observed that germline-related genes were further de-repressed in E7.5 *Dnmt*-null Epi compared to *Dnmt1*-KO and *Dnmt3a/3b*-DKO embryos”.

5. The generation of the *Tet/Dnmt-6KO* is intriguing, but the rationale, phenotypic comparisons and biological insights of these experiments could to be improved (lines 208-239):

• In the absence of 5mC substrate due to the lack of DNMTs in embryogenesis, can the authors explain further what insights the *Tet/Dnmt-6KO* provide about TET function(s)?

Response: We greatly appreciate the reviewer’s suggestions. In fact, DNA methylation patterns and levels show dynamic changes during early development. Specifically, DNMTs and TETs coordinate to regulate the global and site-specific DNA methylation in mouse pre- and post-implantation embryo^{15,16}. However, while it is a well acceptable in the field that the function of DNA methylation is independent of TET mediated DNA demethylation, it has not yet been directly tested experimentally. Meanwhile, we are also curious about the phenotypes of methylation/demethylation free embryo, and whether the phenotype is more severe or better than that of the *Dnmt*-null embryo. Moreover, it is still not clear whether the non-catalytic roles of TET are critical for mouse embryonic development. Therefore, it is important to test it by removal of both *Dnmts* and *Tets* in zygotes. In this study, we established IMGZ system to generate *Tet/Dnmt1-4KO*, *Tet/Dnmt3a/3b-5KO*, and *Tet/Dnmt-6KO* embryos and found that these embryos exhibit similar phenotypes shown in *Dnmt1*-KO, *Dnmt3a/3b*-DKO, and *Dnmt*-null embryos, respectively. Our results provide the most direct evidence to date in support of the opinion that DNA methylation function is independent of TET-mediated DNA demethylation. Our data also suggest that the non-catalytic roles of TET are not essential for mouse embryonic development, since we obtained *Dnmt*-null embryos without 5mC signals where the non-catalytical activity of TET was intact, but the developmental phenotype of *Tet/Dnmt-6KO* embryos did not become more serious. Thus, we have rearranged the description of this section in our revised manuscript.

• The authors compare the phenotypes of *Tet/Dnmt-6KO* and *Dnmt-KO* embryos, concluding that “These [*Tet/Dnmt-6KO*] phenotypes are similar to *Dnmt1/3a/3b-TKO* embryos...” (lines 214-220). Can the authors please provide quantitative comparisons of the embryos and primitive streak formation to support this conclusion?

We concluded that *Dnmt*-null and *Tet/Dnmt-6KO* have similar phenotypes for embryonic development, based on the findings that they have similar normal morphology at the blastocyst stage, and can form normal egg cylinder at E6.5, but failed to expand and form a normal amniotic cavity in E7.5 Epi (Fig. 2e, 4a and Extended Data Fig. 5d, 7c). Besides, both *Dnmt*-null and *Tet/Dnmt-6KO* embryos failed to display headfolds and heart as recognizable as controls at E8.5 (Fig. 2e, 4b). It is great suggestion to add some quantitative comparisons to strengthen this conclusion. Thus, we calculated the ratio of Epi area to the whole embryo. The results showed that both *Dnmt*-null and *Tet/Dnmt-6KO* embryos display normal Epi development at E6.5, but retarded at E7.5 compared to the control embryos (Fig. R3a, b). In addition, we also analyzed the formation of the primitive streak in E7.5 *Tet/Dnmt-6KO* embryos by RNA in situ hybridization of the *T* probe. The results showed that the ectopic distribution of *T* in *Dnmt*-null and *Tet/Dnmt-6KO* embryos is very similar. We calculated the ratio of expressing *T* regions to whole embryos, which was also similar in *Dnmt*-null and *Tet/Dnmt-6KO* (Fig. R3c, d). We have added some of these data to the Extended Data Fig. 7 in the revised manuscript.

Fig. R3 | *Dnmt*-null and *Tet/Dnmt*-6KO embryos show gastrulation failure.

a, Representative images of IMGZ-derived control, *Dnmt*-null, and *Tet/Dnmt*-6KO embryos at E6.5 and E7.5. *Oct4-EGFP* signaling indicated the epiblast region. The embryo region consists of extraembryonic ectoderm and epiblast. Scale bar, 100 μ m. **b**, The ratio of epiblast area to the whole embryo of control, *Dnmt*-null, and *Tet/Dnmt*-6KO embryos at E6.5 and E7.5. Epiblast was indicated by the signal of *Oct4-EGFP*. Data are mean \pm s.e.m of indicated biological replicates. Significance was determined by a two-tailed, unpaired Student's t-test. ** $P < 0.01$; n.s, no significant difference. **c**, Representative images of RNA in situ hybridization of *T* probe in the *Dnmt*-null and *Tet/Dnmt*-6KO embryos at E7.5, showing primitive streak elongation failure in these embryos. More than

three independent embryos were analyzed for each group. Scale bar, 100 μ m. **d**, The ratio of primitive streak area to whole embryo of *Dnmt*-null and *Tet/Dnmt*-6KO embryos at E7.5. The primitive streak was indicated by the signal of *T*. Data are mean \pm s.e.m of indicated biological replicates. Significance was determined by a two-tailed, unpaired Student's t-test. n.s., no significant difference.

• *To properly disentangle the roles and functions of DNMTs and TETs (lines 208-239), it seems essential to include Tet KO embryos. Tet TKO embryos die at gastrulation, so without including these embryos for comparison, I think it cannot be concluded that the Dnmt/Tet-6KO embryos are more like Dnmt-null rather than Tet TKOs. Previous reports demonstrated that loss of Dnmt3a or Dnmt3b in fact partially rescued embryogenesis the Tet TKO (Dai et al. 2016 Nature), so I think there are important biological insights that remain unexplored within this section.*

Thanks for this suggestion. To this, we used IMGZ system to simultaneously inactivate *Tet1/2/3* in the zygotic genome, by introducing premature stop codon at Q1071, Q174, and Q178 amino acid mutations and disrupted the proteins of TET1, TET2, and TET3, respectively (Fig. 1k). Liquid chromatography-mass spectrometry (LC-MS) showed undetectable 5-hydroxymethylcytosine (5hmC) signal in *Tet*-TKO embryos, confirming that these premature stop mutants of TET1/2/3 destroyed the activity of dioxygenases (Fig. 1j). Strikingly, we found that zygotic disruption of TET1/2/3 did not influence embryo gastrulation. *Tet*-TKO embryos can form the primitive streak at E7.5 and reach the early somite stage with recognizable headfolds, heart, and somites at E8.5 (Fig. 1l, m). However, *Tet*-TKO embryos showed retardation at E9.5 and degeneration at E10.5 (Extended Data Fig. 3j). These findings indicate that zygotic inactivation of Tet enzymes leads to mid-gestation embryonic lethality (Extended Data Fig. 4j), which is inconsistent with the previous study showing severe gastrulation failure in conditional germline knock-out of *Tet1/2/3* embryos¹⁷. Interestingly, a recent study shows that removal of TET proteins at E7.5 leads to mid-gestation embryonic lethality¹⁸, a phenotype similar to IMGZ-derived *Tet*-TKO embryos in our study. Given that maternal TET3 is involved in active demethylation in pronucleus of zygotes^{19,20}, these results together indicate that the maternal TET-mediated active demethylation in zygotes is critical for mouse embryo gastrulation. It is an intriguing task in future to investigate the molecular mechanisms of maternal TET-mediated active demethylation that may be involved in gastrulation.

6. *The authors report that Dnmt-TKO shows profound developmental delay with impaired formation of the primitive streak and onset of gastrulation – indeed, this phenotype is much more severe than the mid-gestational lethality seen in the Dnmt1 KO or Dnmt3a/3b DKO (Li et al. 1992 Cell; Okano et al. 1999 Cell), suggesting an importance of DNA methylation very early in post-implantation development. Yet, the authors observe remarkably few changes in the Dnmt TKO mRNA transcription and no evident correlation between loss of DNA methylation and gene expression changes. They report a de-repression of miRNAs, which are almost exclusively from the Dlk1-Dio3 imprinted cluster. The authors conclude that this underpins many of the gene expression and developmental changes seen in the Dnmt TKO. This proposed mechanism needs further exploration because there are several other knockout models where loss of imprinting at the Dlk1-Dio3 locus does not cause early lethality.*

Dnmt1 KO embryos are unable to maintain inherited DNA methylation from the germline, and hence lose imprinted gDMRs, including the IG-DMR at the Dlk1-Dio3 locus (as confirmed in this study –

lines 339-344). DNA methylation at the IG-DMR controls the imprinting of the *Dlk1-Dio3* miRNA cluster (Seitz et al. 2004 *Genome Res*); hence, *Dnmt1* KO embryos fail to repress the paternal allele, including the miRNA cluster (Extended data Fig 15b). Parthenogenetic embryos are generated through the activation of an oocyte; consequently, with two maternal alleles, these embryos would also express the *Dlk1-Dio3* miRNA cluster from both alleles. However, both *Dnmt1* KO and parthenogenetic embryos die at mid-gestation (Li et al. 1992 *Cell*; Surani and Barton. 1983 *Science*). The authors need to reconcile why the de-repression of the *Dlk1-Dio3* miRNA cluster in the context of the *Dnmt1* KO and parthenogenetic embryos does not impact gastrulation, while in the *Dnmt1* TKO it does so profoundly. In particular, given the finding that *Dlk1-Dio3* miRNA upregulation was remarkably similar between *Dnmt1* KO and *Dnmt1*-TKO embryos (lines 335-339, Extended data Fig 15b).

Response: We greatly appreciate the reviewer's concerns. We concur that IG-DMR is involved in regulating the expression of the *Dlk1-Dio3* miRNA cluster and removal of methylation at IG-DMR can increase miRNA expression in *Dnmt1*-KO or parthenogenesis embryos. However, those embryos haven't shown such severe growth-retarded phenotypes as the *Dnmt*-null embryos, raising the reviewer's concerns. Actually, some of our results also show the role of IG-DMR methylation in the repression of miRNA expression, including less up-regulated miRNA in *Dnmt*-null embryos when compared to *Dnmt1*-KO embryos than compared to *Dnmt3a/3b*-DKO embryos (Extended Data Fig. 15c) and significantly reduced expression of *Dlk1-Dio3* miRNAs in DKO-AG-haESCs induced by IG-DMR deletion (Fig. 6e and Extended Data Fig. 17g). Nevertheless, we would mention that, in the current study, through comparing the WGBS data of *Dnmt*-null with *Dnmt1*-KO or *Dnmt3a/3b*-DKO embryos, we identified that promoters are also related to miRNA expression, including the *Dlk1-Dio3* miRNA cluster. Indeed, we provided five independent pieces of evidence to support a possible link between promoter methylation and miRNA expression at *Dlk1-Dio3* locus. First, IG-DMR methylation doesn't control promoter methylation because the promoters sustain hypermethylation in both *Dnmt1*-KO and *Dnmt3a/3b*-DKO embryos (Extended Data Fig. 16a, b) and IG-DMR deletion in AG-haESCs didn't increase the methylation level of the promoters (Extended Data Fig. 17e). Second, in mature oocytes, where DNA methylation was partially maintained at IG-DMR but with high levels at promoters, the majority of miRNA at *Dlk1-Dio3* were not expressed (Extended Data Fig. 17a-d). Third, pre-implantation embryos kept the hypermethylation state at these promoters and sustained a similar suppression state of *Dlk1-Dio3* miRNAs as oocytes (Extended Data Fig. 17c, d). Fourth, IG-DMR deletion that mimics the IG-DMR methylation can't rescue gastrulation phenotypes in *Dnmt*-null SC embryos, suggesting that promoter methylation may be also involved in inhibition of miRNA expression. Fifth, another two HMP-related up-regulated miRNAs, *let-7b-5p* and *miR-296-3p*, were up-regulated in all tested *Dnmt*-null embryos compared to control embryos (Extended Data Fig. 13c, 15b). Finally, after decreasing dosage of six miRNAs in the *Dnmt*-null embryos, the phenotype of primitive streak elongation failure could be partially rescued (Fig. 6j). Together, while we concur that IG-DMR plays the majority role in suppressing *Dlk1-Dio3* miRNAs, our comparison study reveals that promoters are also related to suppression of the miRNAs, which may be involved in fine-tuning the gastrulation. Besides, compared to *Dnmt1*-KO or parthenogenesis embryos, *Dnmt*-null showed methylation-free in the whole genome, which can influence other epigenetic modifications and genome stability and strengthen the function of up-regulated miRNAs. Therefore, we tone down our statements in the revised manuscript.

Comments:

1. *The mechanism that underpins active demethylation during embryogenesis remains contentious, as TET3-mediated demethylation does not appear to be the primary mechanism (Amouroux et al. 2016 Nat Cell Biol), as described in lines 57-61. Please revise the phrasing of this sentence.*

Response: We greatly appreciate the reviewer's suggestions. We have added one sentence of "Notably, de novo DNA methylation by DNMT proteins also occurs amid global demethylation, and is subjected to TET-mediated DNA demethylation in zygote" to describe the NCB's finding.

2. *The statement "these strategies can't exclude the potential influences of the family genes in gametogenesis, and thus can't distinguish the roles of gametogenesis and/or embryogenesis" (lines 71-73) is over-stated. Conventionally, heterozygous constitutive knockout animals are used to derive knockout embryos, and germline conditional knockouts are used to generate gamete-specific knockouts; these strategies robustly differentiate between the effects of a gene in gametogenesis or embryogenesis. Please amend or delete this statement.*

Response: We apologize for this over-statement. We have deleted this description in the revised manuscript.

3. *Can the authors please check the images in Figure 3e – it appears that the images for Dnmt1 KO and the Dnmt-TKO have been switched?*

Response: Thank you very much. We have corrected this mistake in the revised manuscript.

4. *I disagree with the conclusions that are based on the data presented in Fig 4c (lines 226-229). The higher amount of 5mC observed in the Tet/Dnmt3a/3b-5KO than Tet/Dnmt1-4KO E8.5 embryos does not necessarily indicate that methylation established by de novo DNMTs are vulnerable to TETs because the absence of DNMT1 means that any established de novo methylation will not be maintained. Indeed, in Fig 4e, the levels of DNA methylation in the Tet/Dnmt3a/3b-5KO appear similar to those in the Dnmt3a/3b DKO.*

Response: We apologize for this confusion. We agree with the point that the lower 5mC signals in the Tet/Dnmt1-4KO embryos than Tet/Dnmt3a/3b-5KO does not necessarily indicate that methylation established by de novo DNMTs are vulnerable to TETs. In fact, the original LC-MS results of Dnmt3a/3b-DKO and Tet/Dnmt3a/3b-5KO were not from the same batch of experiments. To exclude the batch effect and compare the differences between them more rigorously, we re-performed the LC-MS analysis of these mutants. As reviewer 3 suggested, we calculated the methylation levels using 5mC over dG instead of dC. The results showed that the 5mC signals in the Dnmt1-KO and Dnmt3a/3b-DKO embryos appear similar to those in the Tet/Dnmt1-4KO and Tet/Dnmt3a/3b-5KO (0.49% vs 0.51% and 1.10% vs 1.04%), respectively, indicating that the DNA methylation protected by DNMT1 or DNMT3A/3B is independent of TET proteins (Fig. 2d, 4c). Thus, we amended this description in the revised manuscript.

5. *The phrasing around the findings presented in Extended data Fig 8a, "... Dnmt1-KO samples were much closer to late development stages of control embryos" (lines 249-250) is somewhat misleading. The Dnmt1-KO embryos cannot maintain DNA methylation, so the apparent 'similarity' between Dnmt1-KO embryos and E6.5/E7.5 embryos is a transient one, which will be lost once the*

de novo DNMTs are no longer expressed. Please consider revising this statement.

Response: We apologize for this confusion. We found that the WGBS data of E6.5 and E7.5 *Dnmt3a/3b*-DKO samples were much closer to control blastocysts while *Dnmt1*-KO samples were much closer to late developmental stages of control embryos (Extended Data Fig. 8a). These results suggested that the DNMT1 can sustainably maintain DNA methylation remained in blastocyst to the late-stage embryo, so *Dnmt3a/3b*-DKO samples were more similar to control blastocysts. The DNMT3A/3B can de novo DNA methylation on the basis of methylation remained in the blastocysts, so *Dnmt1*-KO samples were much closer to the late developmental stages of control embryos. To uniform these findings, we have changed this description to “Interestingly, E6.5 and E7.5 *Dnmt3a/3b*-DKO samples were much closer to control blastocysts, indicating that the DNMT1 can sustainably maintain DNA methylation remained in blastocyst to the late-stage embryo” in the revised manuscript.

6. *Can the authors ensure that the same terminology for the knockouts is used throughout the manuscript? For example, Dnmt-null and Dnmt-TKO are used interchangeably.*

Response: Thank you for this suggestion. We have uniformed this item to *Dnmt*-null in the revised manuscript.

7. *The authors report that Dnmt3a/3b DKO embryos show higher levels of DNA methylation than blastocysts (lines 266-269, Extended data Fig 9), concluding that DNMT1 has de novo methyltransferase function, similar to recent reports (Haggerty et al. 2021 Nature). However, the findings reported in this study are not consistent with that reported by Haggerty et al. The previous study found evidence for de novo activity of DNMT1 at a subset of repetitive elements in the absence of DNMT3A and 3B; whereas, the findings reported here show a significant gain of DNA methylation in Dnmt3a/3b DKO embryos across genomic features. The discussion of these results should be amended.*

Response: Thank you for this suggestion. Haggerty *et al.* reported that DNMT1 can catalyze DNA methylation in both de novo and maintenance contexts, especially at retrotransposons²¹. They found that 50% of the genome gains more than 5% methylation, even in the absence of the *Dnmt3a/3b*, compared to the ICM, pointing out a rather widespread DNMT3-independent de novo methylation activity (Fig. 1b in their paper)²¹. We analyzed the distribution of this 50% hypermethylated site in the genome, and found that 54.5% of the annotated promoters, 53.5% of the enhancers, and 37.5% of the TEs belong to them (Fig. R4a), suggesting that DNMT1 has de novo methylation activity on these genomic elements. In addition, we analyzed the WGBS data of the *Dnmt1*-KO and *Dnmt3a/3b*-DKO E6.5 Epi used in Haggerty’s paper compared to the E4.0 ICM. Consistently, the methylation levels of overlapped hypermethylated retrotransposons ($n = 7837$), enhancers ($n = 1551$), and promoters ($n = 456$) of E6.5 epi revealed in our study were also significantly increased in *Dnmt3a/3b*-DKO samples used in the Haggerty’s paper (Fig. R4b). These results indicated that DNMT1 has de novo methylation activity on multiple genomic elements. Thus, we have added Haggerty’s paper to our references in the revised manuscript.

Fig. R4 | DNMT1 has de novo methylation activity in promoters, enhancers, and TEs

a, Haggerty's paper found that 50% of the genome gains more than 5% methylation, even in the absence of the *Dnmt3a/3b*, compared to the ICM. The barplots showed the percentage of promoters, enhancers, and retrotransposons with more than 50% hypermethylated regions in these 50% genomic regions. Numerators indicate the numbers of hypermethylated elements in *Dnmt3a/3b*-DKO embryos. Denominators indicate the total number of hypermethylated elements in the genome. **b**, Box plots showed methylation levels of overlapped hypermethylated retrotransposons, enhancers, and promoters between *Dnmt1*-KO vs *Dnmt*-null and *Dnmt3a/3b*-DKO vs *Dnmt*-null in Epi at E6.5 shown in Extended Data Fig. 9. The WGBS data of WT, *Dnmt1*-KO, *Dnmt3a*-KO, *Dnmt3b*-KO, and *Dnmt3a/3b*-DKO Epi at E6.5 were obtained from Haggerty's paper. Significances of *Dnmt1*-KO vs E4.0-ICM and *Dnmt3a/3b*-DKO vs E4.0-ICM was determined by Student's unpaired two-sided *t*-test. ****P* < 0.001.

8. Can the authors further explain why there are such dramatic differences in gene expression between the *Dnmt*-TKO E6.5 epiblast versus controls, but not in E7.5 epiblast (Extended data Fig 11e)? Given the dramatic phenotype and lack of effective gastrulation in the *Dnmt*-TKO embryos, the gene expression differences at E7.5 should be substantial.

Response: Thank you for this suggestion. The Extended data Fig 11e in the original manuscript showed the differentiated expression genes (DEGs) related to overlapped hypermethylated enhancers (HMEs) shown in Fig. 5b. There are 1551 HMEs in E6.5 Epi, but only 252 in E7.5 Epi (Fig. 5b). Thus, there are fewer DEGs related to HMEs of E7.5 Epi shown in Extended data Fig 11e. If we analyzed the 953-HMEs related genes of E6.5 Epi in E7.5 samples, there are also 28 up-regulated genes and 30 down-regulated genes in *Dnmt*-TKO Epi (Fig. R5a). The down-regulated genes are also enriched in early embryo development and specification (Fig. R5b). Besides, the RNA-seq results also revealed that there are more DEGs of *Dnmt*-null compared to *Dnmt1*-KO and *Dnmt3a/3b*-DKO in E7.5 Epi than E6.5 Epi (Fig. 3c, e), which is consistent with the dramatic phenotypes in *Dnmt*-null embryos at E7.5.

Fig. R5 | The expression of E6.5 HMEs related genes in E7.5 Epi.

a, Volcano plots show the DEGs (fold change > 2 and adjusted *P*-value < 0.05) of *Dnmt*-null vs control Epi at E7.5 among 953 HME-regulated genes in E6.5 Epi shown in Extended Data Fig. 11d. **b**, Gene ontology enrichments of DEGs related to (a).

9. ESCs can have high levels of genomic instability in 2i conditions. The culture of haploid ESCs in 2i conditions in combination with CRISPR targeting could result in vast genomic instability (lines 360-399). It should be confirmed that the genome of these cells is at least somewhat intact.

Response: Thank you for this suggestion. We agree with the reviewer that ESCs culture in 2i (inhibitors of Mek1/2 and Gsk3β) medium for the long-term can produce genomic instability.

However, we found that the androgenetic haploid embryonic stem cells (AG-haESCs, also termed sperm-like stem cells) that carry only sperm genome, after multiple rounds of genetical manipulations, still maintain a stable genome and can produce alive semi-cloned mice through intracytoplasmic injection AG-haESC into oocytes (ICAHCI)²². One potential reason could be that regular cell sorting, which is used to enrich haploid cells during haploid ESC maintenance, could be beneficial to genome integrity of haploid ESCs. To further confirm the genomic stability of *Mir*-6KO AG-haESCs, we performed karyotyping analysis and whole genome sequencing (WGS) on the diploidized *Mir*-6KO cells. Karyotyping analysis showed that almost all cells sustained a normal karyotype at O48 and *Mir*-6KO cells (Fig. R6a). The copy number variation (CNV) analysis of WGS data showed that *Mir*-6KO carried a small fragment duplication in chromosome 6, while our CRISPR targets are located at chromosome 15 (*let-7b* knock-out) and 12 (*miR-127*, *miR-541*, *miR-369*, *miR-540*, and *miR-409* knock-out) respectively in *Mir*-6KO cells (Fig. R6b), indicating that this duplication may be spontaneously occurred during DNA replication in a subclone. Taken together, the genome of the *Mir*-6KO cells is overall intact.

Fig. R6 | Genomic integrity analysis of *Mir-6KO* AG-haESCs.

a, Representative karyotype image of O48 and *Mir-6KO* AG-haESCs showed normal karyotype in CRISPR-Cas9 targeted cells. 50 chromosome spreads were analyzed for each cell line. **b**, The copy number variation (CNV) analysis of WGS data from O48 and *Mir-6KO* AG-haESCs. The color indicates scores of duplications (red), wild-type (black), and deletions (blue). The total length of the CNVs is 10Mb, including a large duplication at Chr6 (8Mb). These data showed that the genome of these cell lines is at least somewhat intact.

10. Can the authors further explain the use of androgenetic haploid (AH) ESCs to study the *Dlk1-Dio3* imprinted domain? As the miRNA cluster at this locus is expressed from the maternal allele, why would deletion of the IG-DMR on the paternal allele result in down-regulation of the miRNA cluster in AH-ESCs (lines 368-372, Extended data Fig 17g)?

Response: Thank you for this suggestion. The *Dlk1-Dio3* locus is an essential imprinted region for embryonic development and is controlled by intergenic germline-derived (IG)-DMR²³. Deletion of IG-DMR from the maternal chromosome leads to loss of imprinting (LOI) in this locus and causes embryonic lethal after E16, but paternal deletion does not change the imprinting state of the region, leading to normal embryonic development in the resultant mouse, indicating that deletion of the IG-DMR can mimic DMR methylation state²³. The hypermethylated IG-DMR of paternal genome can suppress the *Dlk1-Dio3* miRNAs expression, while maternal IG-DMR is hypomethylated, leading to the derepression of the maternal *Dlk1-Dio3* miRNAs²⁴. The wild-type AG-haESCs that carry only sperm genome were cultured under conditions with 2i, which induce loss of the paternal IG-DMR methylation, leading to the derepression of the paternal *Dlk1-Dio3* miRNAs. Thus, deletion of IG-DMR in the wild-type AG-haESCs can reestablish the paternal imprinting and mimic the DMR methylation, leading to the down-regulation of *Dlk1-Dio3* miRNAs, which is also proved by

our miRNA sequencing (Fig. 6e and Extended Data Fig. 17g).

11. Can the authors add justification for using >1.414 fold change for the miRNA analysis? Can you please also clarify whether the $p<0.05$ cutoff for significance was corrected for multiple comparisons? (lines 1180-1182)

Response: We thank the reviewer for pointing out these issues. The differentially expressed miRNAs were identified by DESeq2 with default parameters²⁵. DESeq2 first normalized miRNA count by calculating size factors, then estimated and shrunk miRNA dispersion, finally fitted Negative Binomial GLM and calculated P -value by Wald statistics. Since there are about 1300 expressed miRNAs (1972 annotated miRNAs in the mouse genome) in our sequencing data, to obtain more differentially expressed miRNAs, we cut off the absolute value of $\log_2(\text{FC})>0.5$, that is, the fold change >1.414 . For the same reason, only miRNAs with P -values less than 0.05 were considered to be differentially expressed miRNAs, which is also applied in our previously reported work²⁶. To solve these confusions, we have amended the description and added the reference in our revised manuscript.

Reviewer #3 (Remarks to the Author):

This is a very exciting study investigating the role of DNA methylation during gastrulation using dCas9-AID base editors to simultaneously inactivate DNMT1, 3a and 3b. Simultaneous inactivation of genes is very difficult with mouse breeding approaches, and in particular for DNA methylation, many confounding effects would make interpretation very difficult.

The authors tested different base editors (BE), and used the most efficient one to induce triple KO embryos for Tet1. The frequency for homozygous premature mutation was quite high, 56%.

Quite extraordinary is that they manage a 6x mutant (all three DNMTs and all three TETs - simultaneously) – the efficiency was 1 in 6 embryos having all 6 genes mutated!

The IMGZ technology offered a number of insights, some confirmation of previous knowledge, some novel. Among the novel findings, they show evidence that DNA methylation of retrotransposons and enhancers may not be critical for gastrulation. Especially the latter is quite interesting and is unexpected, but they find enhancers relevant for organ and tissue development.

The manuscript was exciting from the beginning but got better and better with every figure. They find that DNA methylation primarily suppresses miRNAs (46.7% of targets) and protein coding genes, but a lower fraction (28%). The majority of the 54 miRNA were in the Dlk1-Dio3 region, and they primarily targeted downregulation of genes that were found downregulated upon DNMT KO.

Response: We greatly appreciate the reviewer's recognition of the significance of our work.

Can the authors speculate what would the effects be if they generated catalytic mutants instead of full KOs? In other words, can one separate catalytic from non-catalytic roles for DNMTs and TETs during development?

Response: We greatly appreciate the reviewer's question. So far, most of the previous studies focused on the catalytic activity of DNMTs and TETs during embryo development. Meanwhile, some of reports demonstrate that DNMTs and TETs can contribute to regulating gene expression through non-catalytic activity, raising the possibility that these proteins are capable of playing a distinct and catalytic activity independent function during embryonic development. DNMT1

contains a catalytically inactive mutation, C1226W, which can recruit H3K4 demethylase KDM1A/LSD1 to repress the expression of a set of targeted genes in HCT116 colon cancer cells²⁷. The N-terminal region of DNMT1 can interact with the E-cadherin transcriptional repressor SNAIL1 to suppress the expression of E-cadherin in HCT116 cells²⁸, which were directly associated with WNT/ β -catenin signaling and involved in embryonic development. TET1 can interact with the SIN3A co-repressor complex and suppress the expression of many targeted genes in mouse ESC²⁹. TET2 can recruit HDAC2 and repress transcription of interleukin-6 (*IL-6*) via histone deacetylation during inflammation resolution in innate myeloid cells³⁰. The non-catalytic action of TET3 binds to the paternal transcribed allele of the imprinted gene Small nuclear ribonucleoprotein-associated polypeptide N (*Snrpn*) and prevents terminal differentiation of adult neural stem cells (NSCs)³¹. These references indicated that the non-catalytic activities of DNMT and TET proteins played important roles in differentiated cells.

However, there is little known about the non-catalytic function of DNMTs and TETs protein during embryonic development. Recently, Arand et al. reported that deficiencies of all three Tet enzymes in germinal vesicle oocytes using a morpholino-guided knockdown approach arrested the *Tet*-TKD embryos beyond the 2-cell stage with the most severe phenotype linked to *Tet2* knockdown. They found that *Tet2*-KD alone prevented normal preimplantation development with only a few embryos capable to develop to the morula stage, which can be partially rescued by co-injection of the truncated *Tet2* mRNA (missing the catalytic domain), suggesting an important non-catalytic function of TET2 to preimplantation development³². In this study, we used IMGZ system to generate *Dnmt* and *Tet* mutants by disruption of both the catalytic and non-catalytic activities of these proteins. It is an intriguing question to separate catalytic from non-catalytic roles for DNMTs and TETs during development. However, we believe it is out of the scope of current manuscript.

Comments:

- the authors developed a website to design sgRNAs and predict off-target effects. Is this website publicly available? What algorithm does it use?

Response: Thanks for these suggestions. We have uploaded the local executable program to github (<https://github.com/DiabloRex/BASE-Editor>), with more options to customize and more functionalities. Users can run the algorithm on their computers with the instructions on github. We will continuously update the program using new technologies (such as MAUI for cross-platform execution), and inform users about our website when it became available. For core algorithms, to search the sgRNAs, we are using the genome fasta file and genome annotation file to locate the sequence for PAM selection. Then, those sgRNAs are subjected to Cas-OFFinder program³³ for off-target analysis. Efficiencies are evaluated by using principles of machine learning models³⁴. We have amended the description and added the reference in our revised manuscript.

- in Fig 2f, DNA methylation was measured by mass spec and the value 5mC/C is 15-20%. Normally this is measured over G and in somatic cells (fully methylated genome) this value is 4%, why do the authors measure it over C?

Response: We thank the reviewer for pointing out this issue. We agree that 5mC/dG is a commonly used method to measure methylation levels in the LC-MS results. In fact, we discarded the dG data of the LC-MS results due to the imprecise signals in our previous experiments. To analyze the data more rigorously, we re-performed the LC-MS analysis of these mutant embryos by the IMGZ

system. We calculated the methylation levels using 5mC over dG instead of dC. The results showed that the value of 5mC/dG in the control embryos at E8.5 is about 4%, which is similar to the fully methylated genome. We have amended the figures (Fig. 1j, 2d, and 4c) in our revised manuscript.

- *What is the embryo attrition in these experiments? From untreated to injected with no DNA, from no DNA injection to injection with only the BE (and separately for sgRNA injection), and between injection on BE+sgRNA with and without catalytic activity? The same question for DNMT experiments.*

Response: We apologize for this confusion. The off-target effect of CBE system is sgRNA-independent in previous report and our results. Thus, the control embryos used in this study derived from the injection of BE mRNA without sgRNA. In the original manuscript, we have summarized the major embryo attrition in the Supplementary Table 5. To comprehensively know the embryo attrition in this study, we summarize the approximate number of embryos injected in different experiments including the control embryos in the table below.

Table. R1 Summary of the embryo attrition in this study.

Experimental purpose	Figures in this paper	Embryo attrition (n = injected embryos)
Screening of BEs	Fig. 1 and Extended Data Fig. 1	586
Screening of BE concentration	Extended Data Fig. 1	355
Off-target analysis	Fig. 1 and Extended Data Fig. 2	754
Tet -TKO analysis	Fig. 1 and Extended Data Fig. 4	620
Dnmt1 -KO analysis	Fig. 2 and Extended Data Fig. 5	586
Dnmt3a/3b -DKO analysis	Fig. 2 and Extended Data Fig. 5	660
Dnmt -TKO analysis	Fig. 2 and Extended Data Fig. 5	1060
Tet/Dnmt1 -4KO analysis	Fig. 4 and Extended Data Fig. 7	420
Tet/Dnmt3a/3b -5KO analysis	Fig. 4 and Extended Data Fig. 7	486
Tet/Dnmt -6KO analysis	Fig. 4 and Extended Data Fig. 7	830
Dnmt -TKO rescue	Fig. 6 and Extended Data Fig. 18	1980
Sum		8337

- *Line 162 directs to Suppl Fig 4c – should it be Suppl Fig 4d? Also, the authors write that “...suggesting that DNMT enzymes may be not essential for the egg cylinder formation”- since DNA methylation is measured at E8.5 (complete loss) and DNMT genes are inactivated in zygotes, could there be that there is a leftover DNA methylation at implantation that gives the phenotypes at E7.5? Is there evidence in the literature that also indicated that DNA methylation is not essential for the onset of gastrulation?*

Response: Thank you very much. We have corrected this mistake in the revised manuscript. In fact, we measured the methylation levels of E8.5 embryos through LC-MS (Fig. 2d). The methylation levels of *Dnmt*-null embryos at E6.5 and E7.5 were also measured by WGBS. The *Dnmt*-null embryos showed complete loss of DNA methylation from E6.5 (Fig. 4e). Here, we further measured the DNA methylation levels in *Dnmt*-null blastocysts through immunofluorescent staining of 5mC and 5hmC signals. As expected, there is a leftover DNA methylation in *Dnmt*-null blastocysts, although this is lower than in control embryos (Fig. R7a, b). However, this residual DNA

methylation is gradually diluted and lost through DNA replication-dependent passive demethylation during post-implantation embryonic development, leading to completely lost before E6.5. Therefore, it is unlikely that the gastrulation failure of *Dnmt*-null embryos at E7.5 is due to the leftover DNA methylation at implantation.

Dynamic regulation of DNA methylation by DNA methyltransferases (DNMT1/3A/3B) and Ten-eleven translocation family of dioxygenases (TET1/2/3) is essential for mammalian embryonic development. The *Tet* triple knockout (*Tet*-TKO) embryos with hypermethylated genome displayed gastrulation failure owing to abnormal Nodal signalling, indicating that the stabilization of DNA methylation is important for embryo gastrulation¹⁷. *Dnmt1* or *Dnmt3a/3b* mutants showed embryo lethality at mid-gestation, whereas *Dnmt*-null embryos have not been achieved. In this study, we speculated that the onset of gastrulation is influenced in methylation-free embryos. Thus, we amended the description “*Dnmt*-null embryos were morphologically indistinguishable from control embryos at E6.5, the onset of gastrulation, suggesting that DNMT enzymes may be not essential for the egg cylinder formation” to “*Dnmt*-null embryos were morphologically indistinguishable from control embryos at E6.5, suggesting that DNMT enzymes may be not essential for the egg cylinder formation” in the revised manuscript.

Fig. R7 | DNA methylation level of *Dnmt*-TKO blastocyst.

a, Immunofluorescent images of 5mC (red) and 5hmC (green) staining in control and *Dnmt*-null blastocysts. Scale bar, 100 μ m. **b**, Average fluorescence intensity of 5mC and 5hmC signalling in control and *Dnmt*-null blastocysts. Significance was determined by Student’s unpaired two-sided *t*-test.

- Line 269, yes and this study shows the same (Haggerty, Meissner PMID: 34140676)

Response: We appreciate the reviewer’s suggestion. We have cited this paper in our revised manuscript.

- Line 281 should be “embryos only slightly influenced” instead of “embryos slightly influenced” since it’s a comparison to the above Extended Fig 10c

Response: Thank you for this suggestion. We have amended this description in the revised manuscript.

- Line 325, in “down-regulated genes” you mean protein coding genes, correct?

Response: Yes. The RNA-seq library in our study was constructed by the optimized SMART-seq2³⁵, which is based on the polyA tail of mRNA. Thus, the mapped genes in our RNA-seq data are almost protein-coding genes. We have also checked these down-regulated genes of Fig. 6b in our database, which are definite protein-coding genes.

References:

- 1 Billon, P. *et al.* CRISPR-Mediated Base Editing Enables Efficient Disruption of Eukaryotic Genes through Induction of STOP Codons. *Molecular cell* **67**, 1068-1079 e1064, doi:10.1016/j.molcel.2017.08.008 (2017).
- 2 Zhang, H. *et al.* Simultaneous zygotic inactivation of multiple genes in mouse through CRISPR/Cas9-mediated base editing. *Development* **145**, doi:10.1242/dev.168906 (2018).
- 3 Li, Q. *et al.* CRISPR–Cas9-mediated base-editing screening in mice identifies DND1 amino acids that are critical for primordial germ cell development. *Nature cell biology* **20**, 1315-1325 (2018).
- 4 Wang, X. *et al.* Efficient base editing in methylated regions with a human APOBEC3A-Cas9 fusion. *Nature biotechnology* **36**, 946-949, doi:10.1038/nbt.4198 (2018).
- 5 Hwang, G. H. *et al.* Web-based design and analysis tools for CRISPR base editing. *BMC bioinformatics* **19**, 542, doi:10.1186/s12859-018-2585-4 (2018).
- 6 Aronson, B. E. *et al.* A bipartite element with allele-specific functions safeguards DNA methylation imprints at the Dlk1-Dio3 locus. *Developmental cell* **56**, 3052-3065.e3055, doi:10.1016/j.devcel.2021.10.004 (2021).
- 7 Kawahara, M. *et al.* High-frequency generation of viable mice from engineered bi-maternal embryos. *Nature biotechnology* **25**, 1045-1050, doi:10.1038/nbt1331 (2007).
- 8 Zhong, C. *et al.* CRISPR-Cas9-Mediated Genetic Screening in Mice with Haploid Embryonic Stem Cells Carrying a Guide RNA Library. *Cell Stem Cell* **17**, 221-232, doi:10.1016/j.stem.2015.06.005 (2015).
- 9 Hirasawa, R. *et al.* Maternal and zygotic Dnmt1 are necessary and sufficient for the maintenance of DNA methylation imprints during preimplantation development. *Genes & Development* **22**, 1607-1616, doi:10.1101/gad.1667008 (2008).
- 10 Li, J., Ishii, T., Feinstein, P. & Mombaerts, P. Odorant receptor gene choice is reset by nuclear transfer from mouse olfactory sensory neurons. *Nature* **428**, 393-399, doi:10.1038/nature02433 (2004).
- 11 Eggan, K. *et al.* Mice cloned from olfactory sensory neurons. *Nature* **428**, 44-49, doi:10.1038/nature02375 (2004).
- 12 Zuo, E. *et al.* Cytosine base editor generates substantial off-target single-nucleotide variants in mouse embryos. *Science* **364**, 289-292, doi:10.1126/science.aav9973 (2019).
- 13 Zuo, E. *et al.* GOTI, a method to identify genome-wide off-target effects of genome editing in mouse embryos. *Nature protocols* **15**, 3009-3029, doi:10.1038/s41596-020-0361-1 (2020).
- 14 Bielinska, M., Narita, N. & Wilson, D. B. Distinct roles for visceral endoderm during embryonic mouse development. *The International journal of developmental biology* **43**, 183-205 (1999).
- 15 Chen, S. *et al.* Developmental abnormalities in supporting cell phalangeal processes and cytoskeleton in the Gjb2 knockdown mouse model. *Disease models & mechanisms* **11**,

- doi:10.1242/dmm.033019 (2018).
- 16 Wu, X. & Zhang, Y. TET-mediated active DNA demethylation: mechanism, function and beyond. *Nature reviews. Genetics* **18**, 517-534, doi:10.1038/nrg.2017.33 (2017).
- 17 Dai, H.-Q. *et al.* TET-mediated DNA demethylation controls gastrulation by regulating Lefty–Nodal signalling. *Nature* **538**, 528-532, doi:10.1038/nature20095 (2016).
- 18 Ma, L. *et al.* Tet-mediated DNA demethylation regulates specification of hematopoietic stem and progenitor cells during mammalian embryogenesis. *Science advances* **8**, eabm3470, doi:10.1126/sciadv.abm3470 (2022).
- 19 Gu, T. P. *et al.* The role of Tet3 DNA dioxygenase in epigenetic reprogramming by oocytes. *Nature* **477**, 606-610, doi:10.1038/nature10443 (2011).
- 20 Guo, F. *et al.* Active and passive demethylation of male and female pronuclear DNA in the mammalian zygote. *Cell stem cell* **15**, 447-459, doi:10.1016/j.stem.2014.08.003 (2014).
- 21 Haggerty, C. *et al.* Dnmt1 has de novo activity targeted to transposable elements. *Nature structural & molecular biology* **28**, 594-603, doi:10.1038/s41594-021-00603-8 (2021).
- 22 Yin, Q. *et al.* Dosage effect of multiple genes accounts for multisystem disorder of myotonic dystrophy type 1. *Cell research* **30**, 133-145, doi:10.1038/s41422-019-0264-2 (2020).
- 23 Lin, S. P. *et al.* Asymmetric regulation of imprinting on the maternal and paternal chromosomes at the Dlk1-Gtl2 imprinted cluster on mouse chromosome 12. *Nature genetics* **35**, 97-102, doi:10.1038/ng1233 (2003).
- 24 Seitz, H. *et al.* A large imprinted microRNA gene cluster at the mouse Dlk1-Gtl2 domain. *Genome research* **14**, 1741-1748, doi:10.1101/gr.2743304 (2004).
- 25 Love, M. I., Huber, W. & Anders, S. Moderated estimation of fold change and dispersion for RNA-seq data with DESeq2. *Genome biology* **15**, 550, doi:10.1186/s13059-014-0550-8 (2014).
- 26 Hua, M. *et al.* Identification of small non-coding RNAs as sperm quality biomarkers for in vitro fertilization. *Cell Discov* **5**, 20, doi:10.1038/s41421-019-0087-9 (2019).
- 27 Clements, E. G. *et al.* DNMT1 modulates gene expression without its catalytic activity partially through its interactions with histone-modifying enzymes. *Nucleic acids research* **40**, 4334-4346, doi:10.1093/nar/gks031 (2012).
- 28 Espada, J. *et al.* Regulation of SNAIL1 and E-cadherin function by DNMT1 in a DNA methylation-independent context. *Nucleic acids research* **39**, 9194-9205, doi:10.1093/nar/gkr658 (2011).
- 29 Williams, K. *et al.* TET1 and hydroxymethylcytosine in transcription and DNA methylation fidelity. *Nature* **473**, 343-348, doi:10.1038/nature10066 (2011).
- 30 Zhang, Q. *et al.* Tet2 is required to resolve inflammation by recruiting Hdac2 to specifically repress IL-6. *Nature* **525**, 389-393, doi:10.1038/nature15252 (2015).
- 31 Montalbán-Loro, R. *et al.* TET3 prevents terminal differentiation of adult NSCs by a non-catalytic action at Snrpn. *Nat Commun* **10**, 1726, doi:10.1038/s41467-019-09665-1 (2019).
- 32 Arand, J. *et al.* Tet enzymes are essential for early embryogenesis and completion of embryonic genome activation. *EMBO reports* **23**, e53968, doi:10.15252/embr.202153968 (2022).
- 33 Bae, S., Park, J. & Kim, J. S. Cas-OFFinder: a fast and versatile algorithm that searches for potential off-target sites of Cas9 RNA-guided endonucleases. *Bioinformatics (Oxford, England)* **30**, 1473-1475, doi:10.1093/bioinformatics/btu048 (2014).
- 34 Arbab, M. *et al.* Determinants of Base Editing Outcomes from Target Library Analysis and Machine Learning. *Cell* **182**, 463-480.e430, doi:10.1016/j.cell.2020.05.037 (2020).

35 Peng, G. *et al.* Molecular architecture of lineage allocation and tissue organization in early mouse embryo. *Nature* **572**, 528-532, doi:10.1038/s41586-019-1469-8 (2019).

REVIEWER COMMENTS

Reviewer #2 (Remarks to the Author):

The authors have addressed the majority of reviewer comments and have done an exceptional amount of work in the revision. In particular, the new quantitative comparisons between the Dnmt-null and Tet/Dnmt-6KO phenotypes, with the inclusion of the Tet-TKO, are robust and convincing, demonstrating that DNMTs function independently of TETs.

However, I still find that the titled conclusion that DNA methylation is indispensable/essential for gastrulation does not have sufficient support. Certainly, the growth and development of Dnmt-null embryos is dramatically stunted at gastrulation, but it is an over-statement to highlight this the cause of embryonic lethality.

Gastrulation is the specification of the three germ layers during primitive streak formation in the embryo. In my opinion, the Brachyury (T) staining along the primitive streak in E7.5 embryos (Fig. 2f), while showing a modest impairment in Dnmt-null embryos (Fig. S7f), is insufficient evidence that there is a failure of gastrulation in the Dnmt-null embryos. There are other mechanisms that could lead to an under-developed epiblast (e.g., cell death due to chromosome instability or accumulation of DNA damage). Thus, I think the authors need to provide further lines of evidence to support that gastrulation is impaired in Dnmt-null embryos, or consider revising the conclusions to clarify that the Dnmt-null embryos die at gastrulation but that the exact mechanism remains unclear.

One possible line of investigation that could be explored is further analysis of the RNA-sequencing data. While down-regulation of gastrulation genes in Dnmt-null embryos supports that gastrulation may be impaired; this dataset could be utilised further. Can the authors look at the expression of marker genes for ectoderm, mesoderm and definitive endoderm (e.g., using those reported in Argelaguet et al. 2019 Nature; PMID: 31827285), to see if there is evidence that specific cells/signatures are missing or deficient in the Dnmt-null embryos? If gastrulation is impaired, one might expect a failure to appropriately specify the mesoderm and/or endoderm.

Minor comments:

1. If the suggestions above are incorporated, in the abstract, the statement "Moreover, DNMT1 or DNMT3A/3B are indispensable for gastrulation..." (lines 43-44) needs to be revised to "DNMT1, 3A and 3B are indispensable for gastrulation." I believe this is a typo, as Dnmt1 KO and Dnmt3a/3b DKO embryos do gastrulate and survive to mid-gestation.

2. In the discussion, the authors conclude "Interestingly, we also found that Dnmt3a/3b-DKO displayed significantly increased levels of DNA methylation compared to blastocysts, indicating that DNMT1 has de novo methylation activity at these sites as shown in recent reports." (lines 283-286). I agree that this is one interpretation of those findings, but another may be that in the absence of DNMT3A and DNMT3B, DNMT1 is able to function more efficiently at maintaining inherited DNA methylation. The comparison that would definitively demonstrate a de novo function of DNMT1 would be comparing Dnmt3a/3b-DKO Epi to Dnmt3a/3b -DKO blastocysts. This is more just a comment, but may be worth discussing.

3. The authors state that "Meanwhile, these common hypermethylated elements mediated by DNMT1 or DNMT3A/3B were mainly distributed at Epi (Fig. 5b), implying critical roles of DNA methylation in Epi but not Exe development, consistent with previous observations of DNA methylation dispensable for the growth and survival of the Exe and hypomethylation in human placenta." (lines 286-290). Indeed, the hypomethylated state of the ExE likely explains why there were less DMRs relative to the Epi, but a recent report has demonstrated that DNA methylation is essential for trophoblast development (Andrews et al. 2023 PMID: 36690623). Please consider revising.

4. Can you please add to the Figure legends for Fig. 3d and f, which comparison the Dnmt-null DEGs were taken from for gene ontology analysis (e.g., Dnmt-null versus Dnmt1-KO)?

Reviewer #3 (Remarks to the Author):

The authors have responded to my questions and have addressed my concerns. Regarding the new data they provide in Fig R7, where they test the "leftover DNA methylation". The authors write "this residual DNA methylation is gradually diluted and lost through DNA replication-dependent passive demethylation during post-implantation embryonic development, leading to completely lost before E6.5."

I am intrigued by this data. The Fig 4e WGBS data shows 0% DNA methylation at E7.5.

1. Antibody staining is not linear, we know that, and if we assume that Dnmt-null blastocysts have about half of DNA methylation of the WT blastocysts, say 50%: if one counts around 80-120 cells in the antibody stained blastocysts and if E6.5 have ~600-700 cells, that is 3 cell divisions in total. One would still expect about 5% DNA methylation, not completely zero. Even if we could another cell division that is still at least 2% (as measured by WGBS). My question is: how can DNA methylation reach completely zero% by E6.5 if the antibody staining shows almost normal levels of DNA methylation at the blastocyst stage?

2. "The Dnmt-null embryos showed complete loss of DNA methylation from E6.5 (Fig. 4e)." The figure legend says E7.5, which one is correct?

The point-by-point response to the reviewers' comments

Reviewer #2 (Remarks to the Author):

The authors have addressed the majority of reviewer comments and have done an exceptional amount of work in the revision. In particular, the new quantitative comparisons between the Dnmt-null and Tet/Dnmt-6KO phenotypes, with the inclusion of the Tet-TKO, are robust and convincing, demonstrating that DNMTs function independently of TETs.

However, I still find that the titled conclusion that DNA methylation is indispensable/essential for gastrulation does not have sufficient support. Certainly, the growth and development of Dnmt-null embryos is dramatically stunted at gastrulation, but it is an over-statement to highlight this the cause of embryonic lethality.

Gastrulation is the specification of the three germ layers during primitive streak formation in the embryo. In my opinion, the Brachyury (T) staining along the primitive streak in E7.5 embryos (Fig. 2f), while showing a modest impairment in Dnmt-null embryos (Fig. S7f), is insufficient evidence that there is a failure of gastrulation in the Dnmt-null embryos. There are other mechanisms that could lead to an under-developed epiblast (e.g., cell death due to chromosome instability or accumulation of DNA damage). Thus, I think the authors need to provide further lines of evidence to support that gastrulation is impaired in Dnmt-null embryos, or consider revising the conclusions to clarify that the Dnmt-null embryos die at gastrulation but that the exact mechanism remains unclear.

One possible line of investigation that could be explored is further analysis of the RNA-sequencing data. While down-regulation of gastrulation genes in Dnmt-null embryos supports that gastrulation may be impaired; this dataset could be utilised further. Can the authors look at the expression of marker genes for ectoderm, mesoderm and definitive endoderm (e.g., using those reported in Argelaguet et al. 2019 Nature; PMID: 31827285), to see if there is evidence that specific cells/signatures are missing or deficient in the Dnmt-null embryos? If gastrulation is impaired, one might expect a failure to appropriately specify the mesoderm and/or endoderm.

Response: We appreciate the reviewer's comment. We agree with the reviewer that the under-developed epiblast of the Dnmt-null embryo at E7.5 may be caused by impaired differentiation, cell death, or other mechanisms. In the current study, we have found the impaired primitive streak of Dnmt-null embryo at E7.5 by Brachyury (T) staining, which cannot reach the early somite stage with recognizable headfolds like Dnmt1-KO and Dnmt3a/3b-DKO embryos (Fig. 2e, f), indicating that DNMT1, DNMT3A, and DNMT3B are critical for mouse gastrulation. We have also found that some key signal pathways that control gastrulation and primitive streak formation in mice were mis-regulated in Dnmt-null embryos (Extended Data Fig. 6a). Meanwhile, the marker genes involved in the primitive streak formation and germ layer determination were also disrupted in Dnmt-null embryos (Extended Data Fig. 6b). To further determine whether gastrulation of the Dnmt-null embryo is impaired, as the referee suggested, we analyzed the expression changes of marker genes for ectoderm, mesoderm, and definitive endoderm in Dnmt-null Epi. The results showed that there are many differentiated expressed germ-layer enhancer-related genes for ectoderm, mesoderm, and definitive endoderm in E7.5 Epi of Dnmt-null embryos compared to both Dnmt1-KO and Dnmt3a/3b-DKO embryos, but fewer at E6.5 Epi, indicating that the expression of germ layer specific genes is also impaired by mutation of DNMT1, DNMT3A, and DNMT3B (Extended Data Fig. 7a-c in the revised manuscript). Together, we believed that the loss of DNA methylation in the

genome by inactivation of *Dnmt1//3a/3b* causes aberrant expression of gastrulation-related genes, leading to embryonic lethal during gastrulation. Nevertheless, the exact mechanism remains unclear. We have modified our main text and figures accordingly in the revised manuscript.

Minor comments:

1. If the suggestions above are incorporated, in the abstract, the statement “Moreover, DNMT1 or DNMT3A/3B are indispensable for gastrulation...” (lines 43-44) needs to be revised to “DNMT1, 3A and 3B are indispensable for gastrulation.” I believe this is a typo, as *Dnmt1* KO and *Dnmt3a/3b* DKO embryos do gastrulate and survive to mid-gestation.

Response: Thank you very much. We have amended this description in the revised manuscript.

2. In the discussion, the authors conclude “Interestingly, we also found that *Dnmt3a/3b*-DKO displayed significantly increased levels of DNA methylation compared to blastocysts, indicating that DNMT1 has *de novo* methylation activity at these sites as shown in recent reports.” (lines 283-286). I agree that this is one interpretation of those findings, but another may be that in the absence of DNMT3A and DNMT3B, DNMT1 is able to function more efficiently at maintaining inherited DNA methylation. The comparison that would definitively demonstrate a *de novo* function of DNMT1 would be comparing *Dnmt3a/3b*-DKO Epi to *Dnmt3a/3b* -DKO blastocysts. This is more just a comment, but may be worth discussing.

Response: We appreciate the reviewer’s comment. We agree with the referee’s opinion that DNMT1 may function more efficiently at maintaining inherited DNA methylation during the absence of DNMT3A/3B proteins. Thus, we have discussed and modified the main text accordingly in the revised manuscript.

3. The authors state that “Meanwhile, these common hypermethylated elements mediated by DNMT1 or DNMT3A/3B were mainly distributed at Epi (Fig. 5b), implying critical roles of DNA methylation in Epi but not Exe development, consistent with previous observations of DNA methylation dispensable for the growth and survival of the Exe and hypomethylation in human placenta.” (lines 286-290). Indeed, the hypomethylated state of the ExE likely explains why there were less DMRs relative to the Epi, but a recent report has demonstrated that DNA methylation is essential for trophoblast development (Andrews et al. 2023 PMID: 36690623). Please consider revising.

Response: Thank you for this suggestion. We have rearranged this description in the revised manuscript.

4. Can you please add to the Figure legends for Fig. 3d and f, which comparison the *Dnmt*-null DEGs were taken from for gene ontology analysis (e.g., *Dnmt*-null versus *Dnmt1*-KO)?

Response: Thank you for this suggestion. We have added the description to the legends of the revised manuscript.

Reviewer #3 (Remarks to the Author):

The authors have responded to my questions and have addressed my concerns. Regarding the new data they provide in Fig R7, where they test the “leftover DNA methylation”. The authors write “this residual DNA methylation is gradually diluted and lost through DNA replication-dependent passive

demethylation during post-implantation embryonic development, leading to completely lost before E6.5."

I am intrigued by this data. The Fig 4e WGBS data shows 0% DNA methylation at E7.5.

1. Antibody staining is not linear, we know that, and if we assume that *Dnmt*-null blastocysts have about half of DNA methylation of the WT blastocysts, say 50%: if one counts around 80-120 cells in the antibody stained blastocysts and if E6.5 have ~600-700 cells, that is 3 cell divisions in total. One would still expect about 5% DNA methylation, not completely zero. Even if we could another cell division that is still at least 2% (as measured by WGBS). My question is: how can DNA methylation reach completely zero% by E6.5 if the antibody staining shows almost normal levels of DNA methylation at the blastocyst stage?

Response: Thank you very much for this comment. We apologize for this confusion. In fact, the average methylation levels of *Dnmt*-null embryos shown in Fig. 4e and Fig.5a were not completely zero. There is about 0.9% DNA methylation in *Dnmt*-null embryos at E6.5 (Fig. R1a). The average DNA methylation levels of WT blastocysts at E3.5 were about 20%. The DNA methylation was slightly reduced in *Dnmt*-null blastocysts, as shown in immunofluorescence staining. We assume that *Dnmt*-null blastocysts have about 15% DNA methylation. There are about 40 cells in E3.5 blastocyst and 600 cells in E6.5 embryo, involving 4 cell divisions in total (Fig. R1b). Therefore, the DNA methylation theoretically reaches 0.94% by E6.5 in *Dnmt*-null embryos, which is consistent with our results and the reviewer's opinion. Thus, we have amended our description and added a supplementary table (Supplementary Table S7 in the revised manuscript) about the results of WGBS data.

[redacted]

Fig. R1 | The changes of DNA methylation in *Dnmt*-null embryo.

a, Bar plots represent the average CG methylation level measured by WGBS. Data are mean \pm s.e.m of three biological replicates. The methylation data of E3.5 TE and ICM were from Smith's paper ¹.

b, Timeline of preimplantation and early postimplantation development. The figure was from Kojima's paper ².

2. "The *Dnmt*-null embryos showed complete loss of DNA methylation from E6.5 (Fig. 4e)." The figure legend says E7.5, which one is correct?

Response: We apologize for this confusion. The description should be "The *Dnmt*-null embryos showed complete loss of DNA methylation from E6.5 (Fig. 5a)" in the first point-by-point response.. The WGBS data are definitely E7.5 whole embryos in Fig. 4e, while the WGBS data of E6.5 and E7.5 Epi and Exe are shown in Fig. 5a. We have amended Fig. 4e in the 2nd revised manuscript.

References:

- 1 Smith, Z. D. *et al.* Epigenetic restriction of extraembryonic lineages mirrors the somatic transition to cancer. *Nature* **549**, 543-547, doi:10.1038/nature23891 (2017).
- 2 Kojima, Y., Tam, O. H. & Tam, P. P. Timing of developmental events in the early mouse embryo. *Seminars in cell & developmental biology* **34**, 65-75, doi:10.1016/j.semcdb.2014.06.010 (2014).

REVIEWERS' COMMENTS

Reviewer #2 (Remarks to the Author):

The authors have responded to my questions and addressed my concerns. In particular, I appreciate the extended discussion of the findings, added to page 13.

Reviewer #3 (Remarks to the Author):

The authors have addressed my remaining questions. Congratulations to all authors for this effort, beautiful work.

The point-by-point response to the reviewers' comments

Reviewer #2 (Remarks to the Author):

The authors have responded to my questions and addressed my concerns. In particular, I appreciate the extended discussion of the findings, added to page 13.

Response: We appreciate the reviewer's comment.

Reviewer #3 (Remarks to the Author):

The authors have addressed my remaining questions. Congratulations to all authors for this effort, beautiful work.

Response: We greatly appreciate the reviewer's recognition of the significance of our work.